# Transformers Provably Solve Parity Efficiently with Chain of Thought

**Juno Kim**[1,2][*]  **Taiji Suzuki**[1,2]
[1]Department of Mathematical Informatics, University of Tokyo
[2]Center for Advanced Intelligence Project, RIKEN
[*]junokim@berkeley.edu

## Abstract

This work provides the first theoretical analysis of training transformers to solve complex problems by recursively generating intermediate states, analogous to fine-tuning for chain-of-thought (CoT) reasoning. We consider training a one-layer transformer to solve the fundamental $k$-parity problem, extending the work on RNNs by Wies et al. (2023). We establish three key results: (1) any finite-precision gradient-based algorithm, without intermediate supervision, requires substantial iterations to solve parity with finite samples. (2) In contrast, when intermediate parities are incorporated into the loss function, our model can learn parity in one gradient update when aided by *teacher forcing*, where ground-truth labels of the reasoning chain are provided at each generation step. (3) Even without teacher forcing, where the model must generate CoT chains end-to-end, parity can be learned efficiently if augmented data is employed to internally verify the soundness of intermediate steps. Our findings, supported by numerical experiments, show that task decomposition and stepwise reasoning naturally arise from optimizing transformers with CoT; moreover, self-consistency checking can improve multi-step reasoning ability, aligning with empirical studies of CoT.

## 1 Introduction

Large language models (LLMs) based on the transformer architecture (Vaswani et al., 2017) have achieved astounding success across a variety of natural language processing and machine learning tasks (see e.g. Wan et al., 2024; Minaee et al., 2024; Naveed et al., 2024; Zhao et al., 2024). However, they often struggle when tasked with solving complex reasoning problems, especially in a zero-shot setting without any form of intermediate guidance or supervision (Geva et al., 2021; Rae et al., 2022; Arkoudas, 2023; Wang et al., 2024). These failures are particularly evident in tasks requiring multi-hop reasoning or compounded logical steps (Sakarvadia et al., 2024).

A promising approach to overcome these limitations is chain-of-thought (CoT) reasoning, where the model is prompted or fine-tuned to solve complex tasks step-by-step by explicitly making intermediate reasoning steps to arrive at the desired answers (Wei et al., 2022; Kojima et al., 2022). Since its discovery, CoT reasoning has been shown to significantly enhance the problem-solving capabilities of LLMs while also increasing the interpretability and trustworthiness of the reasoning process, and has spawned numerous prompting techniques (Liu et al., 2023; Qiao et al., 2023) and applications for a variety of downstream tasks including common-sense reasoning, mathematical problem-solving, and symbolic or multi-modal reasoning; see e.g. Zhang et al. (2023b); Yu et al. (2023); Chu et al. (2024) for surveys on CoT. In particular, besides being used as a prompting method, directly training or fine-tuning models to generate CoT has also been shown to significantly improve multi-step reasoning performance (Nye et al., 2021; Wei et al., 2022; Zelikman et al., 2022; Lightman et al., 2024).

Despite these empirical successes, however, the theoretical understanding of the CoT mechanism and task decomposition in transformers is still limited. Existing works focus on characterizing the expressivity of transformers equipped with CoT, providing constructions which can solve certain complexity classes (Feng et al., 2023; Merrill & Sabharwal, 2023; 2024; Li et al., 2024b), studying the class of functions that can be learned in-context with CoT (Li et al., 2023; Bhattamishra et al., 2024), or analyzing the estimation error of multi-step models (Hu et al., 2024). Nevertheless, such

approaches do not indicate how such capabilities might emerge when training transformers to generate reasoning chains. Li et al. (2024a) analyze the training dynamics of a one-layer transformer in an in-context learning setting and show that CoT ability may be acquired; however, they do not consider explicitly training with CoT chains, which is a more difficult problem since the objective depends on the recursive application of the transformer to itself.

In this paper, we seek to formalize the mechanism through which stepwise reasoning emerges in transformers optimized to generate CoT chains. We focus on the specific problem of *bit subset parity* (learning the parity of an unknown subset of $k$ bits from a $d$-bit input), which is known to be impossible to learn end-to-end with any finite-precision gradient-based algorithm in polynomial steps (Shalev-Shwartz et al., 2017; Shamir, 2018). In contrast, Wies et al. (2023) have demonstrated that recurrent neural networks (RNNs) can solve parity efficiently when provided with intermediate supervision. We build on this direction to establish positive optimization guarantees for the transformer architecture. Our object of study is a one-layer transformer incorporating a softmax attention layer, feedforward layer and positional encoding, that is recursively applied to its own output to generate a sequence of intermediate parity computations to arrive at the desired output, analogous to CoT generation. Our contributions are summarized as follows.

- We extend the impossibility result for parity (Theorem 1), which was established only for population gradient descent, to the more realistic finite-sample setting in Theorem 2. We prove that any iterative algorithm with access to an approximate gradient oracle for the end-to-end empirical loss cannot solve a random target parity within a specific polynomial number of steps.

- In contrast, we show that when the loss is summed over all intermediate states, by utilizing *teacher forcing*, a form of process supervision wherein ground-truth intermediate steps are provided during training,[1] our model can learn any parity in a single gradient update (Theorem 5). This shows the benefits of training directly with CoT chains to acquire task decomposition ability.

- We further consider training with CoT generated end-to-end without teacher forcing,[2] and show that parity can still be learned in a logarithmic number of steps if augmented data is employed to check the validity of intermediate steps (Theorem 7), thereby mimicking self-consistency checks often used in CoT reasoning (Zelikman et al., 2022; Wang et al., 2023; Huang et al., 2023a).

- We conduct numerical experiments supporting our findings (Section 4 and Appendix D).

Our results provide theoretical insights into how transformers can naturally and efficiently optimize to perform task decomposition, emphasizing the role of explicit intermediate supervision for complex tasks. Moreover, these findings corroborate recent empirical studies on CoT reasoning demonstrating improved performance through process supervision and internal validation of reasoning chains (Huang et al., 2023a; Tian et al., 2024; Lightman et al., 2024).

## 1.1 RELATED WORKS

**Complexity of transformers.** A line of work aims to understand the effectiveness of CoT from the perspective of complexity theory. Feng et al. (2023) show that autoregressive transformers of constant size can solve basic arithmetic tasks by recursively generating CoT reasoning steps, which is not possible when directly generating the solution; this separation arises because looping the generated outputs back to its inputs increases the 'effective depth' of the model. Works such as Chiang et al. (2023); Merrill & Sabharwal (2023) study the expressivity of fixed-precision transformer architectures in terms of classes of formal languages. Merrill & Sabharwal (2024); Li et al. (2024b) show that CoT reasoning enables recognizing wider language classes, and characterizes the increased expressivity depending on the length of the reasoning chain. Sanford et al. (2024) studies the relation between transformers and massively parallel computation protocols, showing that logarithmic depth suffices to solve multi-hop induction tasks that cannot be efficiently solved by other sequence models.

---

[1]Teacher forcing or process supervision is a training procedure for recurrent models in which the model receives the ground truth output at time $t$ as input at time $t + 1$ during training (Goodfellow et al., 2016, p.377). Many fine-tuning methods with ground-truth CoT chains implement teacher forcing, being more effective than output supervision with chains generated end-to-end (Deng et al., 2023; Tian et al., 2024; Lightman et al., 2024).

[2]Teacher forcing can induce exposure bias where a model is not robust to its own errors. In practice, partial (scheduled or random) teacher forcing methods are used to overcome this issue (Bengio et al., 2015; Goyal et al., 2017; Mihaylova & Martins, 2019).

Additionally, Li et al. (2023); Bhattamishra et al. (2024) study the class of functions that can be learned in context by transformers with CoT from the point of view of in-context learning.

**Optimization and generalization of CoT.** Zhu et al. (2024) study the 'reversal curse' via the training dynamics of a one-layer transformer and shows that the model fails to generalize from $A \to B$, $B \to C$ to $A \to C$ as an argument for the necessity of explicit step-by-step reasoning. Hu et al. (2024) study CoT prompting from a statistical estimation perspective by introducing a multi-step latent variable model for CoT and analyzing its approximation, generalization and prompting-based errors. Notably, Li et al. (2024a) study the training dynamics of a one-layer attention-only transformer model in an in-context learning setting and show that CoT generalization capability can be obtained. However, this does not address the possibility or benefits of training with CoT chains. Lightman et al. (2024) empirically study training LLMs with either process or outcome supervision, showing that the former significantly outperforms the latter when training to solve challenging reasoning tasks.

**Parity and task decomposition.** The difficulty of learning parity without task decomposition is established in Shalev-Shwartz et al. (2017); Shamir (2018). The work most relevant to our paper is Wies et al. (2023), which study task decomposition for parity with classical Elman RNNs. They show that by incorporating intermediate states into the loss function and utilizing teacher forcing, parity can be solved with polynomial iterations and embedding size. Our Theorem 5 extends this positive result to autoregressive transformers, rigorously establishing the benefits of CoT-based training.

## 2 PROBLEM SETUP

**Notation.** We write $[n] := \{1, 2, \cdots, n\}$ for any integer $n$. Scalar operations apply componentwise to vectors, e.g. for $\boldsymbol{z} \in \mathbb{R}^n$ we write $\phi(\boldsymbol{z}) = (\phi(z_1), \cdots, \phi(z_n))^\top$, $\boldsymbol{z}^2 = \boldsymbol{z} \odot \boldsymbol{z} = (z_1^2, \cdots, z_n^2)$ and $|\boldsymbol{z}| = (|z_1|, \cdots, |z_n|)^\top$. The 2-norm is always denoted by $\|\cdot\|$. The multi-linear inner product or contraction of $\boldsymbol{z}_1, \cdots, \boldsymbol{z}_r \in \mathbb{R}^n$ for any $r \in \mathbb{N}$ is denoted as $\langle \boldsymbol{z}_1, \cdots, \boldsymbol{z}_r \rangle := \sum_{i=1}^n z_{1,i} \cdots z_{r,i}$. In particular, $\langle \boldsymbol{z}_1 \rangle = \boldsymbol{z}_1^\top \mathbf{1}_n$ and $\langle \boldsymbol{z}_1, \boldsymbol{z}_2 \rangle = \boldsymbol{z}_1^\top \boldsymbol{z}_2$.

### 2.1 THE PARITY PROBLEM

Let $d \geq k \geq 2$ be integers and let $P$ denote the set of size $k$ subsets of $\{1, \cdots, d\}$ equipped with the uniform distribution. In this paper, we study the $k$-parity problem for $d$-bit inputs $\boldsymbol{x} = (x_j)_{j=1}^d \sim \mathrm{Unif}(\{\pm 1\}^d)$, where the output $y = \prod_{j \in p} x_j$ is determined by the parity of an unknown subset of bits $p \in P$. We abuse notation and identify the set of indices $p$ with the corresponding parity mapping $\boldsymbol{x} \mapsto \prod_{j \in p} x_j$. Given $n$ samples $(\boldsymbol{x}^i, y^i)_{i \in [n]}$, our goal is to predict the parity of any test input.

It is known that parity is fundamentally difficult in the sense that it cannot be solved in polynomial time by any finite-precision gradient-based algorithm, such as neural networks. More precisely, let $\{f_\theta \mid \theta \in \Theta\}$ be any differentiable (w.r.t. $\theta$) parametrized model with polynomially bounded gradients, $\|\nabla f_\theta(\boldsymbol{x})\| = O(\mathrm{poly}(d))$, and define the population loss $\bar{L} = \mathbb{E}_{\boldsymbol{x}} \left[ (y - f_\theta(\boldsymbol{x}))^2 \right]$. We presume access to an $\varepsilon$-*approximate gradient oracle* $\widetilde{\nabla}$ for $L$, which takes any $\theta \in \Theta$ as query and returns a vector $\widetilde{\nabla} \bar{L}(\theta)$ satisfying $\|\widetilde{\nabla} \bar{L}(\theta) - \nabla \bar{L}(\theta)\|_2 \leq \varepsilon$, potentially in an adversarial manner. Then the following holds:

**Theorem 1** (Wies et al. (2023), Theorem 4). *Let $\ell_{0-1}$ be the zero-one loss. There exists an $O(e^{-d/3})$-approximate oracle $\widetilde{\nabla}$ such that[3] the output $\theta(\mathcal{A})$ of any iterative algorithm $\mathcal{A}$ which sequentially makes at most $O(\mathrm{poly}(d))$ queries to $\widetilde{\nabla} \bar{L}$ must satisfy*

$$\mathbb{E}_{\boldsymbol{x}} \left[ \ell_{0-1}(p(\boldsymbol{x}), f_{\theta(\mathcal{A})}(\boldsymbol{x})) \right] \geq \tfrac{1}{2} - O(e^{-d})$$

*with probability at least $1 - O(e^{-d/3})$, when the target parity $p$ is uniformly sampled from $P$.*

The intuition is that the set $P$ of parity functions is exponentially large in the sense that all elements of $P$ are pairwise orthogonal with respect to the data distribution. This implies that the variance of

---

[3]The original paper states that $\mathcal{A}$ can be any iterative gradient-based algorithm which receives an $\Omega(e^{-d/3})$-approximation of the gradient at each step. However, to be more precise, the result is only valid for certain adversarial perturbation schemes.

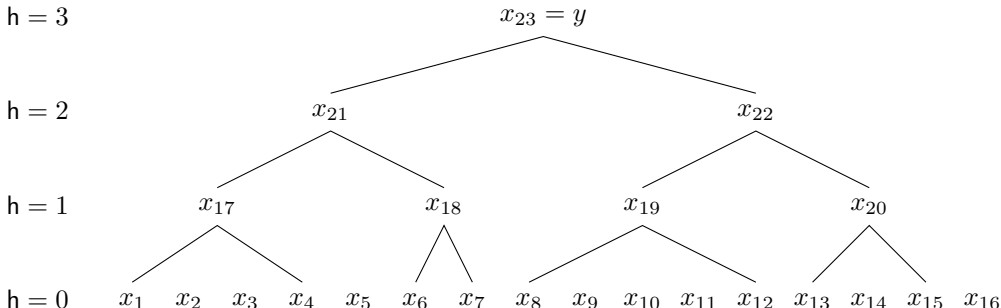

Figure 1: A hierarchical decomposition of an 8-parity problem for $d = 16$. Here $x_{17} = x_1 x_4$ so that $c_1[17] = 1$, $c_2[17] = 4$, $p[17] = 21$ and $h[17] = 1$.

each gradient call $\nabla \bar{L}(\theta)$ with respect to the target parity $p$ is exponentially small (Shalev-Shwartz et al., 2017) and is drowned out by the noise from the adversarial oracle, so that no information can be gained on the target without exponentially many queries. See Section 3.1 for more details.

**Task decomposition.** As in Wies et al. (2023), we assume $k = 2^v$ for an integer $v$ for simplicity and decompose the problem into a hierarchy of 2-parity computations which can be efficiently learned in a sequential manner by our model. This is expressed as a complete binary tree $\mathcal{T}$ of height $v$ and $2k - 1$ nodes. The lowest level contains $k$ nodes representing the bits $x_{j_m}$ for $m \in [k]$. The remaining nodes are labeled $x_{d+1}, \cdots, x_{d+k-1}$ starting from the next lowest level and moving upwards, left to right. The largest index in level $\ell$ for $0 \leq \ell \leq v$ is denoted as $d_\ell = d + \sum_{j=1}^{\ell} 2^{v-j}$, $d_0 = d$. Also, for each $m > d$, the indices of the two child nodes of $x_m$ are denoted as $c_1[m], c_2[m]$ where $1 \leq c_1[m] < c_2[m] < m$. In addition, the parent node index of $x_m$ is denoted as $p[m]$ and the level or height of $x_m$ is denoted as $h[m]$, so that $d_{h[m]-1} < m \leq d_{h[m]}$.

## 2.2 TRANSFORMER MODEL

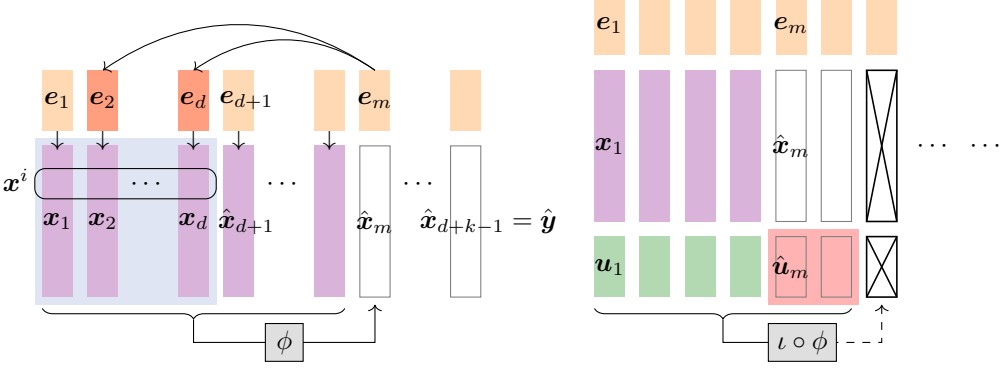

(a) Recursive generation of intermediate states.  (b) Filtered generation with data augmentation.

Figure 2: Illustration of the recursive data generation process by the transformer model. (a) Each token consists of a one-hot positional encoding $e_j$ and parity data $x_j$. The $d$ input tokens (blue) are fixed. The token $\hat{x}_m$ is generated at the $(m - d)$th step by computing attention scores based on position, combining the previous tokens and applying the feedforward layer $\phi$. $\hat{x}_{d+k-1}$ is returned as the model prediction. (b) For the no teacher forcing setup in Section 3.3, data augmentation $u_j$ is implemented to check for self-consistency. If the augmented outputs from the previous generation (red) are uninformative, a filter $\iota$ is applied to zero out the subsequent output.

We study a one-layer transformer architecture employing absolute positional encoding and a single-head softmax attention layer followed by a shallow feedforward layer; skip connections are omitted for simplicity. See Figure 2 for a visualization of our setup.

*Data encoding*: Each input token $\boldsymbol{x}_j = (x_j^i)_{i=1}^n$ for $j \in [d]$ is the $n$-dimensional vector consisting of the $j$th bit of each sample $\boldsymbol{x}^i$. We also add dummy tokens $\boldsymbol{x}_{d+1}, \cdots, \boldsymbol{x}_{d+k-1}$ initially set to $\boldsymbol{0}_n$, which will learn to sequentially generate the actual intermediate nodes. Each $\boldsymbol{x}_j$ is concatenated with the one-hot positional encoding $\boldsymbol{e}_j \in \mathbb{R}^{d+k-1}$ for $j \in [d+k-1]$ to form the internal input $\boldsymbol{p}_j = (\boldsymbol{x}_j^\top \; \boldsymbol{e}_j^\top)^\top \in \mathbb{R}^{n+d+k-1}$ to the attention layer.

*Softmax attention layer*: The attention layer is defined as in (1) in terms of key, query and value matrices $\mathbf{K}, \mathbf{Q}, \mathbf{V}$. We fix the first $n$ columns of $\mathbf{K}, \mathbf{Q}$ to zero so that the attention scores are determined by only the positional encodings. This ensures that the transformer focuses on learning which positions contribute to the parity at each step. $\mathbf{K}, \mathbf{Q}$ are then reparametrized by a single matrix $\mathbf{W} \in \mathbb{R}^{(d+k-1)^2}$; conversely, the value matrix is set to only preserve the $\boldsymbol{x}$ component, as follows.

$$\mathbf{K}^\top \mathbf{Q} = \begin{pmatrix} \boldsymbol{0}_{n \times n} & \boldsymbol{0}_{n \times (d+k-1)} \\ \boldsymbol{0}_{(d+k-1) \times n} & \mathbf{W} \end{pmatrix}, \quad \mathbf{V} = \begin{pmatrix} \mathbf{I}_{n \times n} & \boldsymbol{0}_{n \times (d+k-1)} \end{pmatrix}.$$

This type of reparametrization is common in the literature to make dynamical analysis tractable (Zhang et al., 2023a; Huang et al., 2023b; Mahankali et al., 2023; Kim & Suzuki, 2024).

*Feedforward layer*: The feedforward layer realizes a fixed link function $\phi : [-1, 1] \to [-1, 1]$, applied elementwise and only to the $\boldsymbol{x}_j$ component; the positional encodings are not affected. To exploit the decomposition of our task into 2-parities, we choose $\phi$ such that $\phi(0) = -1$, $\phi(\pm 1) = 1$ so that sums are converted into parities, i.e. $\phi(\frac{a+b}{2}) = ab$ for $a, b \in \{\pm 1\}$. Moreover, we require that $\phi'(0) = \phi'(\pm 1) = 0$ and assume $\phi$ is symmetric and sufficiently regular, so that we may expand $\phi(t) = -1 + ct^2 + O(|t|^4)$ and $\phi'(t) = 2ct + O(|t|^3)$.

The transformer computes $\mathrm{TF}(\boldsymbol{x}_1, \cdots, \boldsymbol{x}_{d+k-1}; \mathbf{W}) = (\hat{\boldsymbol{x}}_1, \cdots, \hat{\boldsymbol{x}}_{d+k-1})$ where the original data $\hat{\boldsymbol{x}}_j = \boldsymbol{x}_j, j \in [d]$ remain unchanged and tokens $\hat{\boldsymbol{x}}_{d+1}, \cdots, \hat{\boldsymbol{x}}_{d+k-1}$ are computed as

$$\hat{\boldsymbol{x}}_m = \phi(\hat{\boldsymbol{z}}_m), \quad \hat{\boldsymbol{z}}_m = \sum_{j=1}^{m-1} \mathbf{V} \hat{\boldsymbol{p}}_j \cdot \mathsf{softmax}(\hat{\boldsymbol{p}}_j^\top \mathbf{K}^\top \mathbf{Q} \hat{\boldsymbol{p}}_m) = \sum_{j=1}^{m-1} \sigma_j(\boldsymbol{w}_m) \boldsymbol{x}_j, \tag{1}$$

where the softmax scores $\sigma_j(\boldsymbol{w}_m) = e^{w_{j,m}} / \sum_{\alpha=1}^{m-1} e^{w_{\alpha,m}}$. Here, we have implicitly added the causal mask $w_{j,m} \leftarrow -\infty$ to the attention layer for $j \geq m$ or $m \leq d$. Note that each $\hat{\boldsymbol{z}}_m, \hat{\boldsymbol{x}}_m$ will be contained in the cube $[-1, 1]^d$ as long as the input tokens are also contained in $[-1, 1]^d$.

**Chain of thought.** Consider repeatedly applying $\mathrm{TF}(\cdot)$ to its own output to generate a 'reasoning chain.' Since the input tokens are fixed, the token $\hat{\boldsymbol{x}}_{d+1}$ will be updated once and then always yield the same value afterwards. Next, since $\hat{\boldsymbol{x}}_{d+2}$ depends on the input tokens and $\hat{\boldsymbol{x}}_{d+1}$, it will be updated twice before becoming fixed. Repeating this, the entire chain stops updating after at most $k - 1$ steps, yielding the output

$$\mathrm{TF}^{(k-1)}(\boldsymbol{x}_1, \cdots, \boldsymbol{x}_d, \boldsymbol{0}_n, \cdots, \boldsymbol{0}_n; \mathbf{W}) = (\hat{\boldsymbol{x}}_1, \cdots, \hat{\boldsymbol{x}}_{d+k-1})$$

where the intermediate predictions are recursively computed as $\hat{\boldsymbol{x}}_m = \phi(\sum_{j=1}^{m-1} \sigma_j(\boldsymbol{w}_m) \hat{\boldsymbol{x}}_j)$. Finally, the top node is returned as the model prediction $\hat{\boldsymbol{y}} = \hat{\boldsymbol{x}}_{d+k-1}$.

This process can be seen as a simplified version of CoT reasoning, albeit not in an in-context learning setting: instead of one-shot predicting $y^i$ from $\boldsymbol{x}^i$, the model starts by solving simpler subtasks and uses the information to attack compound problems, learning to generate intermediate reasoning steps $\boldsymbol{x}_{d+1} \to \cdots \to \boldsymbol{x}_{d+k-1}$ to finally arrive at the desired solution. Importantly, this process is not possible if the model is only trained on the one-shot data $(\boldsymbol{x}^i, y^i)_{i \in [n]}$ as we show in Section 3.1. Instead, we incorporate the prediction error for all intermediate states directly into our loss function (Lightman et al., 2024). We also consider shortening the reasoning chain by using a different causal mask in Section 3.3, which will result in improved control of error and faster convergence.

## 3 MAIN RESULTS

### 3.1 HARDNESS OF PARITY WITHOUT CoT

Before analyzing our transformer model, we first prove a negative learning result in the absence of intermediate supervision that extends Theorem 1, which was stated with respect to the population objective $\bar{L}$ and zero-one test loss $\ell_{0-1}$, to finite samples and mean squared loss.

Let $f_\theta : \{\pm 1\}^d \to \mathbb{R}$ be any differentiable parametrized model and suppose we select the target parity $p$ uniformly at random from $P$. In the finite-sample setting, $n$ i.i.d. samples $(\boldsymbol{x}^i, y^i)_{i \in [n]}$ are generated as $\boldsymbol{x}^i \sim \mathrm{Unif}(\{\pm 1\}^d)$, $y^i = p(\boldsymbol{x}^i)$ and we are given access to (approximate) gradients from the empirical loss

$$L_n(\theta) = \frac{1}{2n} \sum_{i=1}^n (y^i - f_\theta(\boldsymbol{x}^i))^2 = \frac{1}{2} \|p - f_\theta\|_n^2,$$

where $\|\cdot\|_n$ is the empirical norm. It is important that the model $f_\theta$ is applied to each $\boldsymbol{x}^i$ separately and does not cross-reference between different samples, as there exist more efficient parity-learning algorithms if the data is allowed to be manipulated freely. For example, Gaussian elimination can solve parity with $O(d)$ samples and $O(d^3)$ iterations (Raz, 2018). Moreover, this implies that neural networks trained with stochastic gradient descent can also solve parity in polynomial time (Abbe & Sandon, 2020). Instead, in our setting the model is forced to learn from the averaged gradient signal and can only implicitly utilize the correlation between samples.

We show the following result for learning parities with finite-samples in Appendix A:

**Theorem 2** (hardness of finite-sample parity). *Suppose $k = \Theta(d)$.*

(1) *If $n = e^{\Omega(d)}$ and $f_\theta$ has polynomially bounded gradients, there exists an $e^{-\Omega(d)}$-approximate gradient oracle $\widetilde{\nabla}$ such that with probability $1 - e^{-\Omega(d)}$ over random sampling, the output $\theta(\mathcal{A})$ of any iterative (possibly randomized) algorithm which makes at most $O(\mathrm{poly}(d))$ queries to $\widetilde{\nabla} L_n$ has $L_2$-loss lower bounded as*

$$\mathbb{E}_{p \in P, \boldsymbol{x}} \left[ (p(\boldsymbol{x}) - f_{\theta(\mathcal{A})}(\boldsymbol{x}))^2 \right] \geq 1 - e^{-\Omega(d)}.$$

(2) *If $n = \Omega(d^\nu)$ and $\|\nabla f_\theta\| = O(d^{\nu_1})$, there exists an $O(d^{-\nu_2})$-approximate gradient oracle $\widetilde{\nabla}$ such that with probability $1 - e^{-\Omega(d)}$ over random sampling, the output $\theta(\mathcal{A})$ of any iterative (possibly randomized) algorithm which makes at most $O(d^{\nu_3})$ queries to $\widetilde{\nabla} L_n$ has $L_2$-loss lower bounded, where $\nu = 4\nu_1 + 4\nu_2 + 2\nu_3 + 2\nu_4 + 1$, as*

$$\mathbb{E}_{p \in P, \boldsymbol{x}} \left[ (p(\boldsymbol{x}) - f_{\theta(\mathcal{A})}(\boldsymbol{x}))^2 \right] \geq 1 - O(d^{-\nu_4}).$$

We remark that the bounds are asymptotically optimal since $f_\theta \equiv 0$ is a valid estimator. Moreover, the expectation over $p \in P$ can be replaced by the corresponding 'with high probability' statement.

A counter-intuitive aspect of the above result is that parity becomes potentially more difficult when the number of samples increases. Indeed, with exponential samples $n = e^{\Omega(d)}$ (1) we basically recover the statement of Theorem 1, while the guarantees for $n = \mathrm{poly}(d)$ (2) are also polynomial in $d$. This is because the difficulty of parity (Theorem 1) fundamentally depends on the following result:

**Proposition 3** (Shalev-Shwartz et al. (2017), Theorem 1). *Suppose $x$ be a random variable in $\mathbb{R}^d$. Let $\mathcal{H}$ be a class of bounded real-valued functions on $\mathbb{R}^d$ such that $\mathbb{E}_{\boldsymbol{x}}[h(\boldsymbol{x})h'(\boldsymbol{x})] = 0$ for any two distinct $h, h' \in H$ and $f_\theta$ a differentiable parametric model with gradients bounded by $\mathbb{E}_{\boldsymbol{x}}[\|\nabla f_\theta\|^2] \leq F(\theta)^2$. Then for the loss $F_h(\theta) := \mathbb{E}_{\boldsymbol{x}}[(h(\boldsymbol{x}) - f_\theta(\boldsymbol{x}))^2]$ where $h$ is chosen uniformly at random from $\mathcal{H}$, the gradient variance is bounded as*

$$\mathrm{Var}(\theta; \mathcal{H}) := \mathbb{E}_{h \in \mathcal{H}} \left[ \|\nabla F_h(\theta) - \mathbb{E}_{h' \in H}[\nabla F_{h'}(\theta)]\|^2 \right] \leq \frac{F(\theta)^2}{|\mathcal{H}|}.$$

Since all $\binom{d}{k} = e^{\Theta(d)}$ parities in $P$ are pairwise orthogonal with respect to the uniform distribution $\mathrm{Unif}(\{\pm 1\}^d)$, it follows that the variance of $\nabla \bar{L}$ is exponentially small and the target signal can be drowned out by a correspondingly small noise from the oracle. However, this is not true for the empirical distribution which cannot distinguish all elements in $P$ with only $\mathrm{poly}(d)$ samples; the empirical correlation of two random parities will generally be $\Theta(n^{-1/2})$. Therefore a more careful decorrelation argument is needed, resulting in the weaker guarantees of Theorem 2(2). Another technical difference is that Theorem 1 only considers the strong zero-one loss (more formally, their results can be seen to hold for any parity estimator $\hat{p}_{\theta(\mathcal{A})} \in P$ depending on the algorithm output), while we prove the $L_2$ lower bound for any real-valued estimator $f_{\theta(\mathcal{A})}$.

### 3.2 COT WITH TEACHER FORCING

When training with teacher forcing, at each position $d + 1 \leq m \leq d + k - 1$, the ground-truth labels of the preceding intermediate states $\boldsymbol{x}_1, \cdots, \boldsymbol{x}_{m-1}$ are fed into the transformer input to obtain the predictor $\hat{\boldsymbol{x}}_m$ at the $m$th position,

$$\hat{\boldsymbol{x}}_m = \mathrm{TF}(\boldsymbol{x}_1, \cdots, \boldsymbol{x}_{m-1}, \boldsymbol{0}_n, \cdots, \boldsymbol{0}_n; \mathbf{W})_m.$$

The loss function then computes the squared error over all states,

$$L(\mathbf{W}) = \frac{1}{2n} \sum_{m=d+1}^{d+k-1} \|\hat{\boldsymbol{x}}_m - \boldsymbol{x}_m\|^2. \tag{2}$$

Since each sequence of values $\hat{x}_{d+1,i}, \cdots, \hat{x}_{d+k-1,i}$ are generated depending only on the corresponding sample $\boldsymbol{x}^i$ and the parameter matrix $\mathbf{W}$, this can be rewritten in terms of the augmented labels $\bar{y}^i = (x_{d+1}^i, \cdots, x_{d+k-1}^i)^\top$ as

$$L(\mathbf{W}) = \frac{1}{2n} \sum_{i=1}^{n} \|\bar{y}^i - f^\circ(\boldsymbol{x}^i; \mathbf{W})\|^2, \quad f_m^\circ(\boldsymbol{x}^i; \mathbf{W}) = \hat{x}_{m,i}, \quad d + 1 \leq m \leq d + k - 1$$

for a fixed mapping $f^\circ : \{\pm 1\}^d \times \mathbb{R}^{(d+k-1)^2} \to \mathbb{R}^{k-1}$, mirroring the setting of Theorem 2. Hence our model does not cross-reference between samples; moreover, the gradient of $f^\circ$ is bounded as

**Lemma 4.** *For all $\boldsymbol{x}, \mathbf{W}$ it holds uniformly that $\|\nabla_{\mathbf{W}} f^\circ(\boldsymbol{x}; \mathbf{W})\| \leq O(\sqrt{d})$.*

At inference time, test inputs $\boldsymbol{x}_1, \cdots, \boldsymbol{x}_d$ are randomly generated and the prediction for $\boldsymbol{y}_{\text{test}} = p(\boldsymbol{x}_1, \cdots, \boldsymbol{x}_d)$ is computed by iterating TF to generate all $k - 1$ reasoning steps without reference to ground-truth labels; $\hat{\boldsymbol{y}}_{\text{test}} = \mathrm{TF}^{(k-1)}(\boldsymbol{x}_1, \cdots, \boldsymbol{x}_d, \boldsymbol{0}_n, \cdots, \boldsymbol{0}_n; \mathbf{W})_{d+k-1}$. Our positive learning result in this setting is as follows.

**Theorem 5** (CoT with teacher forcing). *Suppose $n = \Omega(d^{2+\epsilon})$ for $\epsilon > 0$, $d$ is sufficiently large and let $\widetilde{\nabla}$ be any $O(d^{-2-\epsilon/8})$-approximate gradient oracle.[4] Set initialization $\mathbf{W}^{(0)} = \boldsymbol{0}$ and learning rate $\eta = \Theta(d^{2+\epsilon/16})$. Then for any target parity $p \in P$, it holds with probability $1 - \exp(-d^{\epsilon/2})$ over random sampling that the one-step update $\mathbf{W}^{(1)} = \mathbf{W}^{(0)} - \eta \widetilde{\nabla} L(\mathbf{W}^{(0)})$ w.r.t. the objective (2) with teacher forcing achieves loss $\|\hat{\boldsymbol{y}}_{\text{test}} - \boldsymbol{y}_{\text{test}}\|_\infty \leq O(d^{-\epsilon/8})$.*

On the other hand, Theorem 2(2) shows that when $n = \Omega(d^{11+\epsilon})$, any iterative algorithm querying an $O(d^{-2-\epsilon/8})$-approximate oracle, with gradients bounded as in Lemma 4, requires more than $\widetilde{\Omega}(d^{\epsilon/4})$ queries to attain a nontrivial ($< \frac{1}{2}$) loss. This establishes a strict separation between learning parities without intermediate supervision and our CoT transformer. The gap increases with more samples as $\epsilon$ increases; moreover, when $n = e^{\Omega(d)}$, we have a much stronger separation by Theorem 2(1), where an exponential number of queries is required to learn $p$.

*Sketch of proof.* The result is shown by explicitly calculating the gradient with respect to each weight $w_{j,m}$ and extracting the gradient signal. As the softmax scores are uniform at initialization, the gradient can be expanded to obtain multilinear contraction or 'interaction' terms between the tokens $\boldsymbol{x}_1, \cdots, \boldsymbol{x}_{m-1}$, one such example being

$$\frac{1}{n} \langle \boldsymbol{x}_m, \hat{\boldsymbol{z}}_m, \hat{\boldsymbol{z}}_m \rangle = \frac{1}{n(m-1)^2} \sum_{\alpha, \beta} \langle \boldsymbol{x}_m, \boldsymbol{x}_\alpha, \boldsymbol{x}_\beta \rangle.$$

In the above equation, if $\alpha, \beta$ are the two child nodes of $m$, the parity $x_\alpha x_\beta x_m \equiv 1$ will be trivial and $\langle \boldsymbol{x}_m, \boldsymbol{x}_\alpha, \boldsymbol{x}_\beta \rangle = n$. On the other hand, for nontrivial parities the interaction strength will generally be $O(\sqrt{n \log d})$ due to sample concentration. For sufficiently large $n$, the trivial parities dominate, allowing us to extract the leading term. Performing these computations up to fourth order interaction terms, we show that the dominating signal of the gradient is $\Theta(d^{-2})$ when $j = c_1[m], c_2[m]$ and $O(d^{-2-\epsilon/8})$ otherwise. Hence the transformer learns to increase only the weights at the relevant positions for each subtask, and is able to compute the desired 2-parity $\hat{\boldsymbol{x}}_m \approx \phi(\frac{1}{2}(\hat{\boldsymbol{x}}_{c_1[m]} + \hat{\boldsymbol{x}}_{c_2[m]})) \approx \hat{\boldsymbol{x}}_{c_1[m]} \hat{\boldsymbol{x}}_{c_2[m]}$ at each node during its forward pass. The full proof is provided in Appendix B. $\square$

---

[4]In fact, we only require that each component of the gradient has error at most $O(d^{-2-\epsilon/8})$ for Theorems 5, 7, which follows since the $L_\infty$ error is bounded above by $L_2$.

## 3.3 CoT without Teacher Forcing

In this section, we extend Theorem 5 to training a transformer without teacher forcing, which is employed alongside teacher forcing in practice to ensure robustness at test time (Bengio et al., 2015; Goyal et al., 2017; Mihaylova & Martins, 2019). The main difficulty in this setting is that wrong answers propagate to later generation steps, exponentially amplifying errors and drowning out the main gradient signals. Error accumulation is also a central practical issue of CoT (Zhang & Parkes, 2023; Wang et al., 2023). To solve this issue, we make some modifications to our transformer model.

First, we minimize the number of required reasoning steps by imposing a slightly stronger form of autoregressivity where each intermediate state $\boldsymbol{x}_m^+$ depends on all tokens $\boldsymbol{x}_j^+$, $j = 1, \cdots, d_{\mathsf{h}[m]-1}$ up to the previous level, rather than the immediately preceding token. This can be expressed as the causal mask $w_{j,m} \leftarrow -\infty$ for $j > d_{\mathsf{h}[m]-1}$ or $m \leq d$; see Figure 3. This ensures that the model gradients are polynomially bounded as in Theorem 2 and that errors can propagate a logarithmic rather than a linear number of steps, and can be easily implemented as the indices $d_\ell$ are known.

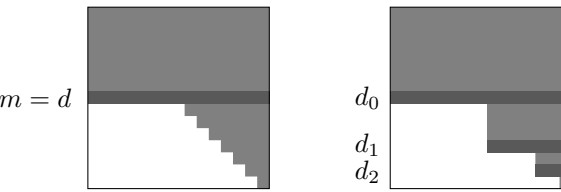

Figure 3: Causal mask for $\mathbf{W}^\top$ with teacher forcing (left); without teacher forcing (right). The gray entries are set to $-\infty$.

Second, we implement a data augmentation technique where random $d$-bit strings $\boldsymbol{u}^i \sim \mathrm{Unif}(\{\pm 1\}^d)$, $i \in [n']$ are appended to the original dataset $(\boldsymbol{x}^i)_{i\in[n]}$. The resulting augmented tokens are denoted as $\boldsymbol{x}_j^+ = (\boldsymbol{x}_j^\top\, \boldsymbol{u}_j^\top)^\top \in \mathbb{R}^{n+n'}$, $\boldsymbol{u}_j = (u_j^i)_{i=1}^{n'}$ so that $\boldsymbol{p}_j = ((\boldsymbol{x}_j^+)^\top\, \boldsymbol{e}_j^\top)^\top$ (the notation is extended to $j > d$), and the key, query and value matrices are appropriately enlarged. The ground truth labels as well as the intermediate states for the augmented data are unknown, so they are not included in the loss function. Nevertheless, unlabeled data can still suffice for self-consistency (Huang et al., 2023a); their purpose is to filter for 'faulty reasoning' in the following sense. If the weights are not sufficiently trained, the output of a node $x_j$ will consist of all nearly $-1$s and thus be uninformative for computing any parities. If the augmented tokens newly generated in the previous iteration of $\mathrm{TF}(\cdot)$ (i.e. up to $\boldsymbol{u}_{d_{\ell-1}}$) are uninformative, we zero out its output on the basis that all subsequent reasoning will be wrong. This is achieved by adding the following filter after the feedforward layer $\phi$:

$$\forall \boldsymbol{z}^+ \in \mathbb{R}^{n+n'}, \quad \iota_\ell(\boldsymbol{z}^+) = \begin{cases} \mathbf{0} & \|\boldsymbol{u}_j + \mathbf{1}_{n'}\|_\infty < \varepsilon_0 \text{ for any } d_{\ell-2} < j \leq d_{\ell-1}, \\ \boldsymbol{z}^+ & \text{otherwise.} \end{cases}$$

Without teacher forcing, during training the entire reasoning chain is generated by iteratively applying TF to its own output until convergence, which takes $v = \log_2 k$ rather than $k - 1$ steps due to the imposed block autoregressivity. Hence $\mathrm{TF}^{(v)}(\boldsymbol{x}_1^+, \cdots, \boldsymbol{x}_d^+, \mathbf{0}_{n+n'}, \cdots; \mathbf{W}) = (\hat{\boldsymbol{x}}_1^+, \cdots, \hat{\boldsymbol{x}}_{d+k-1}^+)$ where the tokens $\hat{\boldsymbol{x}}_{d+1}^+, \cdots, \hat{\boldsymbol{x}}_{d+k-1}^+$ are recursively generated per level as

$$\hat{\boldsymbol{x}}_m^+ = \iota_{\mathsf{h}[m]} \circ \phi(\hat{\boldsymbol{z}}_m^+), \quad \hat{\boldsymbol{z}}_m^+ = \sum_{j=1}^{d_{\mathsf{h}[m]-1}} \sigma_j(\boldsymbol{w}_m)\hat{\boldsymbol{x}}_j^+. \tag{3}$$

The loss is computed against the ground-truth labels as in (2). As before, each sequence of generated states depends only on each sample $\boldsymbol{x}^i$ and the augmented data $\mathbf{U} = (\boldsymbol{u}^i)_{i\in[n']}$, so we may express

$$L(\mathbf{W}, \mathbf{U}) = \frac{1}{2n}\sum_{i=1}^n \|\bar{y}^i - f^\times(\boldsymbol{x}^i; \mathbf{W}, \mathbf{U})\|^2, \quad f_m^\times(\boldsymbol{x}^i; \mathbf{W}, \mathbf{U}) = \hat{x}_{m,i} \tag{4}$$

for a fixed mapping $f^\times$, so that the samples are again not cross-referenced. By considering the propagation of gradients up the chain, the gradient of $f^\times$ can be shown to be bounded as follows.

**Lemma 6.** *For all $\boldsymbol{x}, \mathbf{W}, \mathbf{U}$ we have $\|\nabla_{\mathbf{W}} f^\times(\boldsymbol{x}; \mathbf{W}, \mathbf{U})\| \leq O(d^g)$ where $g = \log_2\|\phi'\|_\infty + 1/2$.*

The exact exponent $g$ depends on the shape of $\phi$. Since $\phi(0) = -1$ and $\phi(1) = 1$, it must hold that $\|\phi'\|_\infty > 2$. Conversely, any such $\|\phi'\|_\infty$ may be achieved by taking $\phi$ to be locally quadratic around $0, \pm 1$ and smoothly joining the curve segments with straight lines of slope $\pm(2 + \epsilon)$. Furthermore, such a link function can be realized by a simple shallow feedforward layer using e.g. $O(1)$ ReQU neurons. Hence $g$ can be taken to be arbitrarily close to $1.5$.

Finally, we implement a simple weight quantization method by rounding each entry of $\mathbf{W}$ to the nearest integer after every update; $\mathbf{W}^{(t+1)} = r[\mathbf{W}^{(t)} - \eta \widetilde{\nabla}_\mathbf{W} L(\mathbf{W}^{(t)}, \mathbf{U})]$, where $r : \mathbb{R} \to \mathbb{Z}$ is the nearest-integer operator. Equivalently, the gradients themselves are quantized. Integer-based quantization methods are widely used in practice to accelerate training and reduce memory usage (Wu et al., 2020; Jacob et al., 2018), and have been successfully implemented in LLMs to facilitate efficient fine-tuning (Dettmers et al., 2022; 2023). In our theoretical setting, quantization also allows us to simplify computations involving propagation of error.

In this setting, we obtain the following learning result.

**Theorem 7** (CoT without teacher forcing). *Suppose $n = \Omega(d^{2+\epsilon})$ for $\epsilon > 0$, $n' = \text{poly}(d)$,[5] $d$ is sufficiently large and let $\widetilde{\nabla}$ be any $O(d^{-2-\epsilon/8})$-approximate gradient oracle. Set $\mathbf{W}^{(0)} = \mathbf{0}$ and $\eta = \Theta(d^{2+\epsilon/16})$. Then for any target parity $p \in P$, it holds with probability $1 - \exp(-d^{(\epsilon \wedge 1)/2})$ over random sampling of (original and augmented) data that the sequence of updates $\mathbf{W}^{(t+1)} = r[\mathbf{W}^{(t)} - \eta \widetilde{\nabla} L(\mathbf{W}^{(t)}, \mathbf{U})]$ w.r.t. the objective (4) without teacher forcing achieves exponentially small loss $\|\hat{\boldsymbol{y}}_\text{test} - \boldsymbol{y}_\text{test}\|_\infty \leq \exp(-\Omega(d^{\epsilon/16}))$ in $\log_2 k$ iterations.*

This gives the same order of separation from Theorem 2(2) as in Section 3.2. Hence transformers can learn parities even without teacher forcing, if the consistency of the chain of reasoning is suitably controlled for. Moreover, our result shows that logarithmic time suffices to learn parity by exploiting the hierarchical decomposition in Figure 1. This extends the circuit complexity result in Merrill & Sabharwal (2024), which states that bounded-depth transformers with a logarithmic number of CoT steps can express problems in log-space; Theorem 7 guarantees that transformers of depth one can *learn by gradient descent* any such function in the exponentially large class $P$.

*Sketch of proof.* The idea is to inductively show that each 2-parity subtask $x_m$ at level $\ell$ will become solved at time $t = \ell$. When $t \leq \ell - 2$, $x_m$ cannot utilize its child nodes $x_{c_1[m]}, x_{c_2[m]}$ since they will also not be optimized, so the weights do not change. At time $\ell - 1$, its child nodes learn to output their parities with high precision, so the objective is approximately equivalent to that of Theorem 5. Then the gradient signal will similarly concentrate on $w_{c_1[m],m}, w_{c_2[m],m}$ and $x_m$ will become solved in the next step. It remains to bound the gradients arising from the loss terms further down the chain $x_{d+1} \to \cdots \to x_{d+k-1}$ (propagation of error), and verify that irrelevant weights $w_{j,m}$ ($p[j] \neq m$) and already optimized weights do not change. The full proof is provided in Appendix C. $\quad\square$

## 4 NUMERICAL EXPERIMENTS

In this section, we present numerical experiments which support and complement our theoretical findings. Compared to the carefully calibrated step sizes and weight updates in Theorems 5 and 7, these experiments study a more realistic training scenario by taking relatively small learning rates and tracking the loss trajectories over a longer period of training. We train one-layer transformers based on the architecture described in Section 2 to solve a random $k$-parity problem with $64$-bit inputs for $k = 8, 16, 32$. Specifically, we implement and compare the following four models.

- **Direct:** $\text{TF}(\cdot)$ is applied to itself $k - 1$ times to generate the reasoning chain end-to-end and the model prediction $\hat{\boldsymbol{y}}$ is directly compared to the ground truth $\boldsymbol{y}$ with the prediction loss $\frac{1}{2n} \|\hat{\boldsymbol{y}} - \boldsymbol{y}\|^2$.
- **CoT:** $\text{TF}(\cdot)$ is applied to itself to generate the reasoning chain end-to-end and the sequence of intermediate states is compared to the ground truth as in (2). Here, we also implement the causal mask in Figure 3 (right) so that only $\log_2 k$ iterations are needed, for additional stability.
- **CoT + teacher forcing:** implements the model in Section 3.2 with teacher forcing.
- **CoT + self-consistency:** implements the model in Section 3.3 with the causal mask in Figure 3 (right) and data augmentation for consistency checks. Weight quantization is omitted.

---

[5]Any polynomial order suffices for the number of augmented data samples.

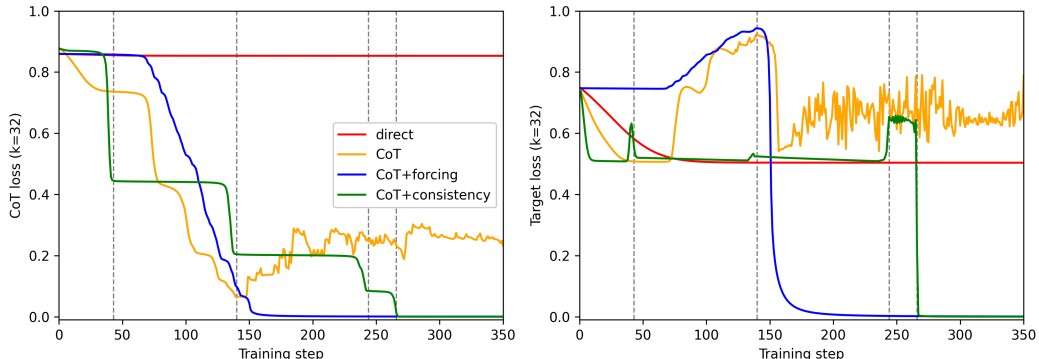

Figure 4: CoT loss (left) and prediction loss (right) curves for the four models when $d = 64$, $k = 32$. For the CoT+consistency model, dashed lines indicate when the filters of each level are deactivated.

All models are optimized using full-batch gradient descent on 100K 64-bit samples with a single Tesla T4 GPU. The three CoT models are trained with the 'CoT loss' (2) scaled by $\frac{1}{k-1}$ to match the prediction loss of the direct model. Figure 4 shows training curves for the CoT loss (left) and the prediction loss (right) over 350 epochs when $k = 32$; results for all $k$ and more details are provided in Appendix D.

We first note that the direct model (red) completely fails to learn the target, plateauing almost immediately. We observed that the weights become nearly uniform so that $\hat{y} \approx \mathbf{0}_n$ and the prediction error is stuck at 0.5. This was not improved by using a multilayer transformer instead of repeated composition. The basic CoT model (yellow) is able to significantly decrease CoT loss but fails to fully solve the problem and eventually becomes unstable. Moreover, the prediction loss never improves beyond 0.5. Indeed, due to the hierarchical structure of parity, the model has no chance of making an informative prediction at the last level $\boldsymbol{x}_{d+k-1}$ unless all preceding levels have been fully solved. In contrast, we verify that CoT with teacher forcing (blue) solves parity efficiently as predicted in Section 3.2, even with a small learning rate. After a burn-in phase, the CoT loss steadily decreases to nearly zero, at which point the prediction loss also decreases rapidly as the final level is solved.

CoT with self-consistency (green) is also able to solve parity efficiently as predicted. Furthermore, the corresponding CoT loss curve clearly exhibits multiple learning stages. In the beginning, the model is essentially optimizing only the first level as subsequent outputs are zeroed out. After a short burn-in phase, the weights are optimized so that the softmax scores concentrate on the relevant nodes, at which point the CoT loss sharply decreases and the filters for the next level are deactivated, unlocking the next learning stage. This phased optimization repeats until all levels are fully solved and is crucial to arriving at the correct answer (in essence, teacher forcing is doing this at all levels simultaneously). Notably, a similar behavior seems to arise in the basic CoT model as well but fails due to accumulating error, further justifying the use of the filtering mechanism.

These results confirm that training explicitly for CoT generation can improve performance on multi-step tasks, and that controlling error accumulation via teacher forcing or self-consistency is key to ensuring proper step-by-step learning.

## 5 CONCLUSION

In this paper, by focusing on the $k$-parity problem, we provide an initial theoretical foundation for training transformers with CoT to perform stepwise reasoning. Our results show that gradient-based learning of parity requires significant iterations without intermediate supervision, but task decomposition using teacher forcing enables efficient learning in a single gradient update. Furthermore, when transformers are trained to generate reasoning chains end-to-end, data augmentation and self-consistency checks can enhance their ability to solve complex tasks. Our work takes the first steps towards understanding how CoT can be leveraged to improve multi-step reasoning capability of foundation models.

ACKNOWLEDGMENTS

JK was partially supported by JST CREST (JPMJCR2015). TS was partially supported by JSPS KAKENHI (24K02905, 20H00576) and JST CREST (JPMJCR2115).

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

# APPENDIX

## A  PROOF OF THEOREM 2

Denote the empirical inner product on $\mathbb{R}^d$ by $\langle f, g \rangle_n = \frac{1}{n} \sum_{i=1}^{n} f(\boldsymbol{x}^i) g(\boldsymbol{x}^i)$ and the corresponding norm as $\|f\|_n^2 = \langle f, f \rangle_n$. We also write

$$L_{n,p}(\theta) = \frac{1}{2} \|p - f_\theta\|_n^2 = \frac{1}{2n} \sum_{i=1}^{n} (p(\boldsymbol{x}^i) - f_\theta(\boldsymbol{x}^i))^2$$

to emphasize the dependency of $L_n$ on $p$. Note that $(d/k)^k \leq \binom{d}{k} \leq (ed/k)^k$ so that $|P| = e^{\Theta(d)}$.

**Bounding gradient variance.** Consider the variance of the empirical gradient $\nabla L_{n,p}$ w.r.t. the target parity $p$:

$$\mathrm{Var}_n(\theta; P) := \mathbb{E}_{p \in P} \left[ \|\nabla L_{n,p}(\theta) - \mathbb{E}_{p' \in P}[\nabla L_{n,p'}(\theta)]\|^2 \right].$$

We proceed to evaluate the magnitude of $\mathrm{Var}_n(\theta; P)$. For $p, p' \in P$ with $p \neq p'$ it holds that

$$\langle p, p' \rangle_n = \frac{1}{n} \sum_{i=1}^{n} \left( \prod_{j \in p} x_j^i \prod_{j' \in p'} x_{j'}^i \right) = \frac{1}{n} \sum_{i=1}^{n} \left( \prod_{j \in p \Delta p'} x_j^i \right).$$

Since $\prod_{j \in p \Delta p'} x_j^i$ is i.i.d. $\mathrm{Unif}(\{\pm 1\})$ for fixed $p, p'$, by applying a union bound over Hoeffding's inequality, it follows for $\delta := \sqrt{4d/n}$ that

$$\mathrm{Pr}\left( \sup_{p \neq p'} |\langle p, p' \rangle_n| \geq \delta \right) \leq |P|(|P| - 1) \exp\left( -\frac{n\delta^2}{2} \right) \leq \binom{d}{k}^2 e^{-2d} \leq \left( \frac{2}{e} \right)^{2d}.$$

Then with probability at least $1 - e^{-\Omega(d)}$ over random sampling, every off-diagonal component of the Gram matrix $G_P := (\langle p, p' \rangle_n)_{p,p' \in P}$ has magnitude at most $\delta$, while the diagonal entries are equal to 1. By the Gershgorin circle theorem, the maximum eigenvalue of $G_P$ satisfies

$$|\lambda_{\max}(G_P) - 1| \leq (|P| - 1)\delta,$$

thus $\lambda_{\max}(G_P) \leq 2(1 \vee |P|\delta)$. This implies that $P$ constitutes a partial frame for the empirical $L^2$ norm with the corresponding frame upper bound. More specifically, for $f : \mathbb{R}^d \to \mathbb{R}$, decompose

$$f = \sum_{p \in P} c_p \cdot p + f_0, \quad f_0 \in (\mathrm{span}\, P)^\perp,$$

for some coefficient sequence $c = (c_p)_{p \in P}$. It follows that

$$\|f\|_n^2 \geq \|f - f_0\|_n^2 = \sum_{p,p' \in P} c_p c_{p'} \langle p, p' \rangle_n = \|G_P^{1/2} c\|^2$$

and

$$\sum_{p \in P} \langle f, p \rangle_n^2 = \sum_{p \in P} \left( \sum_{p' \in P} c_{p'} \langle p, p' \rangle \right)^2 = \|G_P c\|^2 \leq \lambda_{\max}(G_P) \|f\|_n^2.$$

Denoting $D = \dim \Theta$, we can therefore bound $\mathrm{Var}_n(\theta; P)$ as

$$\mathrm{Var}_n(\theta; P) = \inf_{\mu \in \mathbb{R}^D} \mathbb{E}_{p \in P} \left[ \|\nabla L_{n,p}(\theta) - \mu\|^2 \right]$$

$$\leq \mathbb{E}_{p \in P} \left[ \left\| \frac{1}{n} \sum_{i=1}^{n} (f_\theta(\boldsymbol{x}^i) - p(\boldsymbol{x}^i)) \nabla f_\theta(\boldsymbol{x}^i) - \frac{1}{n} \sum_{i=1}^{n} f_\theta(\boldsymbol{x}^i) \nabla f_\theta(\boldsymbol{x}^i) \right\|^2 \right]$$

$$= \mathbb{E}_{p \in P} \left[ \sum_{j=1}^{D} \langle \nabla_{\theta_j} f_\theta, p \rangle_n^2 \right] = \frac{1}{|P|} \sum_{p \in P} \sum_{j=1}^{D} \langle \nabla_{\theta_j} f_\theta, p \rangle_n^2$$

$$\leq \sum_{j=1}^{D} \frac{\lambda_{\max}(G_P)}{|P|} \|\nabla_{\theta_j} f_\theta\|_n^2$$

$$\leq 2 \left( \frac{1}{|P|} \vee \sqrt{\frac{4d}{n}} \right) \sup_{\theta, \boldsymbol{x}} \|\nabla f_\theta(\boldsymbol{x})\|^2.$$

Now by Chebyshev's inequality, for any $\varepsilon > 0$ it holds that

$$\Pr\left( \|\nabla L_{n,p}(\theta) - \mathbb{E}_{p' \in P}[\nabla L_{n,p'}(\theta)]\| > \varepsilon \right) \leq \frac{\mathrm{Var}_n(\theta; P)}{\varepsilon^2}.$$

**Constructing the oracle.** As in Shamir (2018), we define the $\varepsilon$-approximate oracle $\widetilde{\nabla}$ as

$$\widetilde{\nabla} L_{n,p}(\theta) = \begin{cases} \mathbb{E}_{p' \in P}[\nabla L_{n,p'}(\theta)] & \|\nabla L_{n,p}(\theta) - \mathbb{E}_{p' \in P}[\nabla L_{n,p'}(\theta)]\| \leq \varepsilon, \\ \nabla L_{n,p}(\theta) & \text{otherwise.} \end{cases}$$

By union bounding, we see that during $T$ steps the oracle always defaults to the mean gradient and does not reveal any information on the true parity $p$, with probability at least

$$\Pr(Q) \geq 1 - \frac{2T}{\varepsilon^2} \left( \frac{1}{|P|} \vee \sqrt{\frac{4d}{n}} \right) \sup_{\theta, \boldsymbol{x}} \|\nabla f_\theta(\boldsymbol{x})\|^2,$$

where $Q \subseteq P$ denotes the corresponding subset of the hypothesis space. Note that the argument can be extended to any randomized algorithm and random initialization in a straightforward manner by lifting to the product probability space, and so we consider $Q$ to be fixed. Then for any target parity $p \in Q$, the output $\theta(\mathcal{A})$ of the algorithm after $T$ steps does not depend on $p$, so the predictor $f = f_{\theta(\mathcal{A})}$ is also fixed.

**Lower bounding the loss.** We first remark that a simpler proof can be given for the sup norm error, which is enough to establish a separation. Consider arbitrary $p, p' \in P$ with $p \neq p'$ and let $\boldsymbol{x} \in \{\pm 1\}^d$ be such that $p(\boldsymbol{x}) \neq p'(\boldsymbol{x})$, then

$$|p(\boldsymbol{x}) - f(\boldsymbol{x})| + |p'(\boldsymbol{x}) - f(\boldsymbol{x})| \geq |1 - f(\boldsymbol{x})| + |-1 - f(\boldsymbol{x})| \geq 2.$$

Now let $\sigma : Q \to Q$ be any automorphism of $Q$ with no fixed points. The $L_\infty$ error can be bounded below by restricting to the noninformative set $Q$ as follows.

$$\begin{aligned}
\mathbb{E}_{p \in P} \left[ \sup_{\boldsymbol{x}} |p(\boldsymbol{x}) - f_{\theta(\mathcal{A})}(\boldsymbol{x})| \right] &\geq \mathbb{E}_{p \in P} \left[ 1_{\{p \in Q\}} \sup_{\boldsymbol{x}} |p(\boldsymbol{x}) - f(\boldsymbol{x})| \right] \\
&= \frac{1}{2|P|} \sum_{p \in Q} \left( \sup_{\boldsymbol{x}} |p(\boldsymbol{x}) - f(\boldsymbol{x})| + \sup_{\boldsymbol{x}} |\sigma \circ p(\boldsymbol{x}) - f(\boldsymbol{x})| \right) \\
&\geq \frac{1}{2|P|} \cdot 2|Q| = \Pr(Q).
\end{aligned}$$

For mean squared error, we similarly restrict to $Q$ so that

$$\mathbb{E}_{p \in P, \boldsymbol{x}} \left[ (p(\boldsymbol{x}) - f_{\theta(\mathcal{A})}(\boldsymbol{x}))^2 \right] \geq \mathbb{E}_{p \in P, \boldsymbol{x}} \left[ 1_{\{p \in Q\}} (p(\boldsymbol{x}) - f(\boldsymbol{x}))^2 \right].$$

Since the range of $p$ is contained in $[-1, 1]$, the above loss will not increase when $f$ is replaced by its clipped version $\bar{f}(\boldsymbol{x}) = (f(\boldsymbol{x}) \wedge 1) \vee (-1)$. Moreover, in Lemma 8 (proved at the end of the section) we show that $|\mathbb{E}_{p \in P}[p(\boldsymbol{x})]| \leq e^{-\Omega(d)}$ holds with probability $1 - e^{-\Omega(d)}$ over the sample space of $\boldsymbol{x}$, so that

$$\left| \mathbb{E}_{p \in P, \boldsymbol{x}} \left[ p(\boldsymbol{x}) \bar{f}(\boldsymbol{x}) \right] \right| \leq (1 - e^{-\Omega(d)}) \mathbb{E}_{p \in P} \left[ p(\boldsymbol{x}) \right] + e^{-\Omega(d)} \leq e^{-\Omega(d)}$$

and also

$$\begin{aligned}
\mathbb{E}_{p \in P, \boldsymbol{x}} \left[ 1_{\{p \in Q\}} p(\boldsymbol{x}) \bar{f}(\boldsymbol{x}) \right] &= \mathbb{E}_{p \in P, \boldsymbol{x}} \left[ p(\boldsymbol{x}) \bar{f}(\boldsymbol{x}) \right] - \mathbb{E}_{p \in P, \boldsymbol{x}} \left[ 1_{\{p \notin Q\}} p(\boldsymbol{x}) \bar{f}(\boldsymbol{x}) \right] \\
&\leq e^{-\Omega(d)} + (1 - \Pr(Q)) \mathbb{E}_{\boldsymbol{x}} \left[ |\bar{f}(\boldsymbol{x})| \right] \\
&\leq e^{-\Omega(d)} + \frac{(1 - \Pr(Q))^2}{2 \Pr(Q)} + \frac{\Pr(Q)}{2} \mathbb{E}_{\boldsymbol{x}} \left[ \bar{f}(\boldsymbol{x})^2 \right].
\end{aligned}$$

Therefore we may bound

$$
\begin{aligned}
\mathbb{E}_{p\in P,\boldsymbol{x}}\left[(p(\boldsymbol{x})-f_{\theta(\mathcal{A})}(\boldsymbol{x}))^2\right] &\geq \mathbb{E}_{p\in P,\boldsymbol{x}}\left[1_{\{p\in Q\}}(p(\boldsymbol{x})-\bar{f}(\boldsymbol{x}))^2\right] \\
&= \Pr(Q) - 2\mathbb{E}_{p\in P,\boldsymbol{x}}\left[1_{\{p\in Q\}}p(\boldsymbol{x})\bar{f}(\boldsymbol{x})\right] + \Pr(Q)\cdot\mathbb{E}_{\boldsymbol{x}}\left[\bar{f}(\boldsymbol{x})^2\right] \\
&\geq \Pr(Q) - \frac{(1-\Pr(Q))^2}{\Pr(Q)} - 2e^{-\Omega(d)} \\
&\geq 2 - \frac{1}{\Pr(Q)} - 2e^{-\Omega(d)} \\
&\geq 1 - \frac{4T}{\varepsilon^2}\left(\frac{1}{|P|}\vee\sqrt{\frac{4d}{n}}\right)\sup_{\theta,\boldsymbol{x}}\|\nabla f_\theta(\boldsymbol{x})\|^2 - 2e^{-\Omega(d)},
\end{aligned}
\tag{5}
$$

where we have used the inequality $2-(1-t)^{-1}\geq 1-2t$, valid for $t\in[0,\frac{1}{2}]$.

The proof is completed by evaluating the following cases.

(1) If $n=e^{\Omega(d)}$ and $T,\|\nabla f_\theta\|=O(\mathrm{poly}(d))$, the gradient variance is bounded as $\mathrm{Var}_n(\theta;P)\leq e^{-\Omega(d)}$. By taking $\varepsilon=\mathrm{Var}_n(\theta;P)^{1/3}$, it follows that $\Pr(Q)=1-e^{-\Omega(d)}$ and (5) yields the lower bound $1-e^{-\Omega(d)}$.

(2) If $n=\Omega(d^\nu)$, $\|\nabla f_\theta\|=O(d^{\nu_1})$, $\varepsilon=\Theta(d^{-\nu_2})$ and $T=O(d^{\nu_3})$, the gradient variance is bounded as $\mathrm{Var}_n(\theta;P)\leq O(d^{2\nu_1+\nu_3+1/2-\nu/2})=O(d^{-2\nu_2-\nu_4})$ and (5) yields the lower bound $1-O(d^{-\nu_4})$.

$\square$

**Lemma 8.** *If $k=\Theta(d)$, it holds with probability at least $1-e^{-\Omega(d)}$ over random sampling that*

$$
|\mathbb{E}_{p\in P}[p(\boldsymbol{x})]|\leq e^{-\Omega(d)}.
$$

*Proof.* Let $m$ denote the number of $-1$s in $\boldsymbol{x}$. By the Chernoff bound for the binomial distribution,

$$
\Pr\left(\left|m-\frac{d}{2}\right|\leq\frac{\delta d}{2}\right)\geq 1-2\exp\left(-\frac{\delta^2 d}{6}\right)
$$

for a constant $\delta\in(0,1)$ to be determined, so we assume the above event throughout the proof. Moreover denoting the complement parity $p^c=[d]\setminus p$, it holds that $p(\boldsymbol{x})=x_1\cdots x_d\cdot p^c(\boldsymbol{x})$ and $|\mathbb{E}_{p\in P}[p(\boldsymbol{x})]|=|\mathbb{E}_{p\in P}[p^c(\boldsymbol{x})]|$, so it suffices to consider the case where $2k\leq d$.

Without loss of generality, we may assume that $\boldsymbol{x}=(-1,\cdots,-1,1,\cdots,1)$ so that $p(\boldsymbol{x})$ is decided as $(-1)^{|p\cap[m]|}$. We bound the cardinality of the set $P_+:=\{p\in P\mid p(\boldsymbol{x})=1\}$. Each parity in $P_+$ can be determined by choosing $2j$ elements from $[m]$ and $k-2j$ elements from $[d]\setminus[m]$. Denoting by $[t]_j$ the *coefficient* of operator of order $j$, we can evaluate

$$
\begin{aligned}
|P_+| &= \sum_{j=0}^{\lfloor m/2\rfloor}\binom{m}{2j}\binom{d-m}{k-2j} \\
&= \sum_{j=0}^{\lfloor m/2\rfloor}\binom{m}{2j}[t]_{k-2j}(1+t)^{d-m} = \sum_{j=0}^{\lfloor m/2\rfloor}\binom{m}{2j}[t]_k(1+t)^{d-m}t^{2j} \\
&= [t]_k(1+t)^{d-m}\sum_{j=0}^{\lfloor m/2\rfloor}\binom{m}{2j}t^{2j} = \frac{1}{2}[t]_k(1+t)^{d-m}((1+t)^m+(1-t)^m) \\
&= \frac{1}{2}\binom{d}{k}+\frac{1}{2}[t]_k(1-t^2)^{m'}(1+st)^{d-2m'} \\
&= \frac{1}{2}\binom{d}{k}+\frac{s^k}{2}\sum_{j=0}^{\lfloor k/2\rfloor}(-1)^j\binom{m'}{j}\binom{d-2m'}{k-2j},
\end{aligned}
$$

where $m' = m \wedge (d - m)$ and $s = \pm 1$. It further follows that

$$
\left| \frac{|P_+|}{|P|} - \frac{1}{2} \right| \leq \frac{1}{2|P|} \sum_{j=0}^{\lfloor k/2 \rfloor} \binom{m'}{j} \binom{d - 2m'}{k - 2j} \leq \frac{1}{2|P|} \sum_{j=0}^{\lfloor k/2 \rfloor} \binom{\lfloor d/2 \rfloor}{j} \binom{\lfloor \delta d \rfloor}{k - 2j}
$$

$$
\leq \frac{\lfloor k/2 \rfloor}{2} \binom{d}{k}^{-1} \binom{\lfloor d/2 \rfloor}{\lfloor k/2 \rfloor} \binom{\lfloor \delta d \rfloor}{\lfloor \delta d/2 \rfloor} \leq \frac{d}{4} \binom{d - \lfloor d/2 \rfloor - \lfloor \delta d \rfloor}{k - \lfloor k/2 \rfloor - \lfloor \delta d/2 \rfloor}^{-1}
$$

$$
\leq \frac{d}{4} \binom{\lfloor d/4 \rfloor}{\lfloor k/4 \rfloor}^{-1} \leq \frac{d}{4} \left( \frac{d}{k} \right)^{-k/4} = e^{-\Theta(d)}.
$$

Here, we have chosen $\delta = \frac{1}{4} \wedge \frac{k}{2d} = \Theta(1)$ and used the inequality $\binom{a_1 + a_2 + a_3}{b_1 + b_2 + b_3} \geq \binom{a_1}{b_1} \binom{a_2}{b_2} \binom{a_3}{b_3}$. From this, we conclude that

$$
|\mathbb{E}_{p \in P}[p(\boldsymbol{x})]| = \left| \frac{|P \setminus P_+| - |P_+|}{|P|} \right| \leq e^{-\Omega(d)}
$$

with probability $1 - e^{-\Omega(d)}$. $\qquad \square$

## B  Proof of Theorem 5

**Proof of Lemma 4.**  For each $d + 1 \leq m \leq d + k - 1$ and $1 \leq j < m$, the only component of $f^\circ$ depending on $w_{j,m}$ is $f_m^\circ$ and

$$
\left| \frac{\partial f_m^\circ(\boldsymbol{x}; \mathbf{W})}{\partial w_{j,m}} \right| = |\phi'(\hat{z}_m)| \cdot \left| \frac{\partial \hat{z}_m}{\partial w_{j,m}} \right|
$$

$$
\leq \|\phi'\|_\infty \left| \frac{\partial \sigma_j(\boldsymbol{w}_m)}{\partial w_{j,m}} x_j + \sum_{\alpha \neq j} \frac{\partial \sigma_\alpha(\boldsymbol{w}_m)}{\partial w_{j,m}} x_\alpha \right|
$$

$$
= \|\phi'\|_\infty \left| \sigma_j(\boldsymbol{w}_m)(1 - \sigma_j(\boldsymbol{w}_m))x_j - \sigma_j(\boldsymbol{w}_m) \sum_{\alpha \neq j} \sigma_\alpha(\boldsymbol{w}_m) x_\alpha \right|
$$

$$
\leq \|\phi'\|_\infty \sigma_j(\boldsymbol{w}_m)(1 - \sigma_j(\boldsymbol{w}_m)) + \|\phi'\|_\infty \sigma_j(\boldsymbol{w}_m) \sum_{\alpha \neq j} \sigma_\alpha(\boldsymbol{w}_m)
$$

$$
\leq 2\|\phi'\|_\infty \sigma_j(\boldsymbol{w}_m).
$$

Hence it follows that

$$
\sum_{m=d+1}^{d+k-1} \|\nabla_{\mathbf{W}} f_m^\circ\|^2 \leq 4\|\phi'\|_\infty^2 \sum_{m=d+1}^{d+k-1} \sum_{j=1}^{m-1} \sigma_j(\boldsymbol{w}_m)^2 \leq 4\|\phi'\|_\infty^2 (k - 1) = O(d),
$$

as desired. $\qquad \square$

We say that a parity $x_{j_1} \cdots x_{j_r}$ for $1 \leq j_1, \cdots, j_r \leq d + k - 1$ is *trivial* if it always equals 1, or equivalently if its reduction to the independent bits $x_1, \cdots, x_d$ cancel out mod 2. For example, the parity $x_1 x_4 x_{17}$ in Figure 1 is trivial. Define $I_{r,m}$ as the set of nontrivial index $r$-tuples less than $m$:

$$
I_{r,m} = \{(j_1, \cdots, j_r) \mid 1 \leq j_1, \cdots, j_r \leq m - 1, \ x_{j_1} \cdots x_{j_r} \not\equiv 1\}.
$$

In particular, $I_{1,m} = [m - 1]$ since no single parity is trivial.

**Lemma 9** (concentration of interaction terms). *If each bit $x_j^i$ for $i \in [n]$, $j \in [d]$ is i.i.d. generated from the uniform distribution on $\{\pm 1\}$, for any $p > 0$ it holds with probability at least $1 - p$ that*

$$
\max_{\substack{1 \leq r \leq 4 \\ (j_1, \cdots, j_r) \in I_{r,m}}} \frac{|\langle \boldsymbol{x}_{j_1}, \cdots, \boldsymbol{x}_{j_r} \rangle|}{n} \leq \kappa := \sqrt{\frac{2}{n} \log \frac{32d^4}{p}}.
$$

*Proof.* Each tuple $(j_1, \cdots, j_r) \in I_{r,m}$ computes a specific nontrivial parity $x_{j_1} \cdots x_{j_r}$ for which the bits $x_{j_1}^i \cdots x_{j_r}^i$, $i = 1, \cdots, n$ are i.i.d. $\mathrm{Unif}(\{\pm 1\})$ due to symmetry. By Hoeffding's inequality we have that

$$\Pr\left(|\langle \boldsymbol{x}_{j_1}, \cdots, \boldsymbol{x}_{j_r} \rangle| \geq \lambda\right) \leq 2e^{-\lambda^2/2n}.$$

Moreover, $|I_{r,m}| \leq (d+k-1)^r \leq (2d-1)^r$ so that

$$|I_{1,m}| + \cdots + |I_{4,m}| \leq (2d-1) + \cdots + (2d-1)^4 < (2d)^4.$$

Therefore it follows by union bounding that

$$\Pr\left(\max_{1 \leq r \leq 4, (j_1, \cdots, j_r) \in I_{r,m}} |\langle \boldsymbol{x}_{j_1}, \cdots, \boldsymbol{x}_{j_r} \rangle| \geq \lambda\right) \leq 32d^4 e^{-\lambda^2/2n},$$

which implies the statement. $\qquad\square$

In particular, we take $n = \Omega(d^{2+\epsilon})$ and $p = \exp(-d^{\epsilon/2})$ so that $\kappa = O(d^{-1-\epsilon/4})$. This will ensure that the informative gradient signals will dominate the irrelevant interaction terms.

We now proceed to the main proof of Theorem 5. The superscript $(0)$ at initialization is omitted for simplicity. The loss can be written more explicitly as

$$L(\mathbf{W}) = \frac{1}{2n} \sum_{m=d+1}^{d+k-1} \|\phi(\hat{\boldsymbol{z}}_m) - \boldsymbol{x}_m\|^2, \quad \hat{\boldsymbol{z}}_m = \sum_{j=1}^{m-1} \sigma_j(\boldsymbol{w}_m)\boldsymbol{x}_j.$$

It is straightforward to verify for $1 \leq \alpha < m$ that

$$\frac{\partial \sigma_\alpha(\boldsymbol{w}_m)}{\partial w_{j,m}} = (\delta_{j\alpha} - \sigma_\alpha(\boldsymbol{w}_m))\sigma_j(\boldsymbol{w}_m) = (\delta_{j\alpha} - \sigma_j(\boldsymbol{w}_m))\sigma_\alpha(\boldsymbol{w}_m)$$

and

$$\frac{\partial \hat{\boldsymbol{z}}_m}{\partial w_{j,m}} = \sum_{\alpha=1}^{m-1} (\delta_{j\alpha} - \sigma_j(\boldsymbol{w}_m))\sigma_\alpha(\boldsymbol{w}_m)\boldsymbol{x}_\alpha = \sigma_j(\boldsymbol{w}_m)(\boldsymbol{x}_j - \hat{\boldsymbol{z}}_m).$$

Then the gradient of $L$ with respect to each element $w_{j,m}$ at initialization can be computed as

$$\frac{\partial L}{\partial w_{j,m}}(\mathbf{W}) = \frac{1}{n}(\phi(\hat{\boldsymbol{z}}_m) - \boldsymbol{x}_m)^\top \frac{\partial \phi(\hat{\boldsymbol{z}}_m)}{\partial w_{j,m}}$$

$$= \frac{\sigma_j(\boldsymbol{w}_m)}{n}\langle \phi(\hat{\boldsymbol{z}}_m) - \boldsymbol{x}_m, \phi'(\hat{\boldsymbol{z}}_m), \boldsymbol{x}_j - \hat{\boldsymbol{z}}_m \rangle \tag{6}$$

$$= -\frac{1}{n(m-1)}\langle \boldsymbol{x}_m, 2c\hat{\boldsymbol{z}}_m, \boldsymbol{x}_j - \hat{\boldsymbol{z}}_m \rangle \tag{7}$$

$$+ \frac{1}{n(m-1)}\langle -\mathbf{1}_n + c\hat{\boldsymbol{z}}_m^2, 2c\hat{\boldsymbol{z}}_m, \boldsymbol{x}_j - \hat{\boldsymbol{z}}_m \rangle \tag{8}$$

$$+ \frac{1}{n(m-1)}\langle O(|\hat{\boldsymbol{z}}_m|^4), 2c\hat{\boldsymbol{z}}_m, \boldsymbol{x}_j - \hat{\boldsymbol{z}}_m \rangle \tag{9}$$

$$+ \frac{1}{n(m-1)}\langle \phi(\hat{\boldsymbol{z}}_m) - \boldsymbol{x}_m, O(|\hat{\boldsymbol{z}}_m|^3), \boldsymbol{x}_j - \hat{\boldsymbol{z}}_m \rangle. \tag{10}$$

**Computing interaction strengths.** The term (7) will be shown to contain the dominating gradient signal when $j = \mathsf{c}_1[m], \mathsf{c}_2[m]$, while the other terms can be bounded as perturbations. Let $\ell = \mathsf{h}_2[m]$ so that $x_m$ computes a $2^\ell$-parity.

For term (7), we substitute $\hat{\boldsymbol{z}}_m = \frac{1}{m-1}\sum_\alpha \boldsymbol{x}_\alpha$ at initialization to expand

$$\frac{1}{n}\langle \boldsymbol{x}_m, \hat{\boldsymbol{z}}_m, \boldsymbol{x}_j - \hat{\boldsymbol{z}}_m \rangle = \frac{1}{n(m-1)}\sum_\alpha \langle \boldsymbol{x}_m, \boldsymbol{x}_\alpha, \boldsymbol{x}_j \rangle - \frac{1}{n(m-1)^2}\sum_{\alpha,\beta} \langle \boldsymbol{x}_m, \boldsymbol{x}_\alpha, \boldsymbol{x}_\beta \rangle,$$

where the dummy indices $\alpha, \beta, \cdots$ are taken to run over $[m-1]$. Let us evaluate the third-order interaction terms $\langle \boldsymbol{x}_m, \boldsymbol{x}_\alpha, \boldsymbol{x}_\beta \rangle$. If $\mathsf{h}[\alpha] = \ell$, $x_m x_\alpha$ computes the parity of $2^{\ell+1}$ independent bits from $x_1, \cdots, x_d$ so $x_m x_\alpha x_\beta$ cannot be trivial, hence $(m, \alpha, \beta) \in I_{3,m}$ and $|\langle \boldsymbol{x}_m, \boldsymbol{x}_\alpha, \boldsymbol{x}_\beta \rangle| \leq n\kappa$ by Lemma 9. Similarly, $\mathsf{h}[\beta] = \ell$ implies that $(m, \alpha, \beta) \in I_{3,m}$. Suppose $\mathsf{h}[\alpha], \mathsf{h}[\beta] \leq \ell - 1$; unless $\mathsf{h}[\alpha] = \mathsf{h}[\beta] = \ell - 1$, the combined parity $x_\alpha x_\beta$ will not contain enough independent bits to cancel out the $2^\ell$ bits in $x_m$, so again $(m, \alpha, \beta) \in I_{3,m}$. Moreover if $\mathsf{h}[\alpha] = \mathsf{h}[\beta] = \ell - 1$, $x_m x_\alpha x_\beta$ will be trivial if and only if $\{\alpha, \beta\} = \{\mathsf{c}_1[m], \mathsf{c}_2[m]\}$, in which case $\langle \boldsymbol{x}_m, \boldsymbol{x}_\alpha, \boldsymbol{x}_\beta \rangle = n$. Thus we have that

$$\frac{1}{n} \sum_\alpha \langle \boldsymbol{x}_m, \boldsymbol{x}_\alpha, \boldsymbol{x}_\beta \rangle = 2 + \frac{1}{n} \sum_{(m,\alpha,\beta) \in I_{3,m}} \langle \boldsymbol{x}_m, \boldsymbol{x}_\alpha, \boldsymbol{x}_\beta \rangle = 2 + O((m-1)^2 \kappa).$$

Similarly, the contraction $\langle \boldsymbol{x}_m, \boldsymbol{x}_\alpha, \boldsymbol{x}_j \rangle$ can be nontrivial only if $\mathsf{p}[j] = m$ and only when $\alpha$ is the other child node of $x_m$, so that

$$\frac{1}{n} \sum_\alpha \langle \boldsymbol{x}_m, \boldsymbol{x}_\alpha, \boldsymbol{x}_j \rangle = \begin{cases} 1 + O((m-1)\kappa) & \mathsf{p}[j] = m, \\ O((m-1)\kappa) & \text{otherwise.} \end{cases}$$

Since $\kappa = O(d^{-1-\epsilon/4})$ and $d < m \leq 2d - 1$, we can therefore isolate the leading term of order $\Theta(d^{-2})$ as

$$-\frac{1}{n(m-1)} \langle \boldsymbol{x}_m, 2c\hat{\boldsymbol{z}}_m, \boldsymbol{x}_j - \hat{\boldsymbol{z}}_m \rangle$$

$$= -\frac{2c}{(m-1)^2} (1_{\{\mathsf{p}[j]=m\}} + O(d\kappa)) + \frac{2c}{(m-1)^3} (2 + O(d^2\kappa))$$

$$= -\frac{2c}{(m-1)^2} 1_{\{\mathsf{p}[j]=m\}} + O(d^{-2-\epsilon/4}).$$

Next, for term (8), we expand

$$\frac{1}{n} \langle -\mathbf{1}_n + c\hat{\boldsymbol{z}}_m^2, 2c\hat{\boldsymbol{z}}_m, \boldsymbol{x}_j - \hat{\boldsymbol{z}}_m \rangle = -\frac{2c}{n} \langle \hat{\boldsymbol{z}}_m, \boldsymbol{x}_j \rangle + \frac{2c}{n} \langle \hat{\boldsymbol{z}}_m^2 \rangle + \frac{2c^2}{n} \langle \hat{\boldsymbol{z}}_m^3, \boldsymbol{x}_j \rangle - \frac{2c^2}{n} \langle \hat{\boldsymbol{z}}_m^4 \rangle.$$

The second-order terms can be computed as

$$\frac{1}{n} \langle \hat{\boldsymbol{z}}_m, \boldsymbol{x}_j \rangle = \frac{1}{n(m-1)} \left( \langle \boldsymbol{x}_j, \boldsymbol{x}_j \rangle + \sum_{\alpha \neq j} \langle \boldsymbol{x}_\alpha, \boldsymbol{x}_j \rangle \right) = \frac{1}{m-1} + O(\kappa),$$

$$\frac{1}{n} \langle \hat{\boldsymbol{z}}_m^2 \rangle = \frac{1}{n(m-1)^2} \left( \sum_\alpha \langle \boldsymbol{x}_\alpha, \boldsymbol{x}_\alpha \rangle + \sum_{\alpha \neq \beta} \langle \boldsymbol{x}_\alpha, \boldsymbol{x}_\beta \rangle \right) = \frac{1}{m-1} + O(\kappa).$$

We evaluate the fourth-order interaction terms by looking at when $(\alpha, \beta, \gamma, \delta) \notin I_{4,m}$ can occur. Without loss of generality, suppose $\mathsf{h}[\alpha] \leq \mathsf{h}[\beta] \leq \mathsf{h}[\gamma] \leq \mathsf{h}[\delta]$.

   (i) If $\mathsf{h}[\beta] < \mathsf{h}[\gamma] < \mathsf{h}[\delta]$, the parities of $x_\alpha, x_\beta, x_\gamma$ must combine without overlaps to cancel out $x_\delta$, so it must hold that $x_\gamma$ is a child of $x_\delta$ and $x_\alpha, x_\beta$ are the two children of the other child. This subtree is fully determined by the choice of the index $\delta$ and one of its child nodes, so there are at most $O(d)$ trivial 4-tuples in this case.
  (ii) If $\mathsf{h}[\beta] = \mathsf{h}[\gamma] < \mathsf{h}[\delta]$, it still must hold that $\mathsf{h}[\gamma] = \mathsf{h}[\delta] - 1$. Moreover, both $x_\beta, x_\gamma$ must be children of $x_\delta$; otherwise, the bits of $x_\delta$ and the non-child node cannot be canceled out by the remaining nodes. Then either $x_\beta = x_\gamma$ or $x_\beta x_\gamma = x_\delta$, and in both cases we see that $x_\alpha x_\beta x_\gamma x_\delta$ cannot be trivial.
 (iii) If $\mathsf{h}[\beta] < \mathsf{h}[\gamma] = \mathsf{h}[\delta]$, it must be that $\gamma = \delta$, otherwise the bits of $x_\gamma x_\delta$ cannot be canceled out by $x_\alpha x_\beta$. It follows that $x_\alpha x_\beta \equiv 1$ and $\alpha = \beta$, so there are $O(d^2)$ trivial 4-tuples in this case.
 (iv) If $\mathsf{h}[\beta] = \mathsf{h}[\gamma] = \mathsf{h}[\delta]$, it must again hold that two indices must be equal, and the remaining two indices must also be equal, so there are also $O(d^2)$ trivial 4-tuples.

Hence it follows that

$$\frac{1}{n} \langle \hat{\boldsymbol{z}}_m^4 \rangle = \frac{1}{n(m-1)^4} \sum_{\alpha,\beta,\gamma,\delta} \langle \boldsymbol{x}_\alpha, \boldsymbol{x}_\beta, \boldsymbol{x}_\gamma, \boldsymbol{x}_\delta \rangle$$

$$= \frac{1}{n(m-1)^4} \sum_{(\alpha,\beta,\gamma,\delta) \notin I_{4,m}} n + \frac{1}{n(m-1)^4} \sum_{(\alpha,\beta,\gamma,\delta) \in I_{4,m}} O(n\kappa)$$

$$= \frac{|[m-1]^4 \setminus I_{4,m}|}{(m-1)^4} + \frac{|I_{4,m}|}{(m-1)^4} O(\kappa) = O(d^{-2} + \kappa).$$

Furthermore, suppose $\alpha, \beta, \gamma, \delta$ are constrained to contain the index $j$. Then case (i) above counts $O(1)$ nontrivial tuples, while case (i), while cases (iii),(iv) count at most $O(d)$ tuples since there is only one free index to be determined. Hence we also have

$$\frac{1}{n} \left\langle \hat{z}_m^3, x_j \right\rangle = \frac{1}{n(m-1)^3} \sum_{\alpha,\beta,\gamma} \left\langle x_\alpha, x_\beta, x_\gamma, x_j \right\rangle = \frac{O(d)}{(m-1)^3} + O(\kappa) = O(d^{-2} + \kappa).$$

Combining the above, we obtain that

$$\frac{1}{n(m-1)} \left\langle -\mathbf{1}_n + c\hat{z}_m^2, 2c\hat{z}_m, x_j - \hat{z}_m \right\rangle = -\frac{2c}{(m-1)^2} + \frac{2c}{(m-1)^2} + \frac{O(\kappa)}{m-1} = O(d^{-2-\epsilon/4}).$$

For term (9), we note that $\left\langle |\hat{z}_m|^4 \right\rangle = \left\langle \hat{z}_m^4 \right\rangle = O(nd^{-2} + n\kappa)$ as derived above. Then since each component of $\hat{z}_m, x_j - \hat{z}_m$ are contained in $[-1,1], [-2,2]$, respectively, we have that

$$\frac{1}{n(m-1)} \left\langle O(|\hat{z}_m|^4), 2c\hat{z}_m, x_j - \hat{z}_m \right\rangle = \frac{4c}{n(m-1)} O(\langle |\hat{z}_m|^4 \rangle) = O(d^{-2-\epsilon/4}).$$

Finally for term (10), by the Cauchy-Schwarz inequality we have

$$\frac{1}{n} \left\langle |\hat{z}_m|^3 \right\rangle = \frac{1}{n} \sum_{i=1}^n |\hat{z}_{m,i}|^3$$

$$\leq \frac{1}{n} \left( \sum_{i=1}^n \hat{z}_{m,i}^2 \right)^{1/2} \left( \sum_{i=1}^n \hat{z}_{m,i}^4 \right)^{1/2} = \frac{1}{n} \left\langle \hat{z}_m^2 \right\rangle^{1/2} \left\langle \hat{z}_m^4 \right\rangle^{1/2}$$

$$= \frac{1}{n} O(nd^{-1})^{1/2} \cdot O(nd^{-2} + n\kappa)^{1/2} = O(d^{-1-\epsilon/8}),$$

and so we may bound

$$\frac{1}{n(m-1)} \left\langle \phi(\hat{z}_m) - x_m, O(|\hat{z}_m|^3), x_j - \hat{z}_m \right\rangle = \frac{4}{n(m-1)} O(\langle |\hat{z}_m|^3 \rangle) = O(d^{-2-\epsilon/8}).$$

From (7)-(10) we conclude that

$$\frac{\partial L}{\partial w_{j,m}}(\mathbf{W}) = -\frac{2c}{(m-1)^2} \mathbf{1}_{\{\mathsf{p}[j]=m\}} + O(d^{-2-\epsilon/8}),$$

and the same result applies to the approximate gradient $\widetilde{\nabla}_{w_{j,m}} L$ at initialization since the cutoff does not apply and each component of the noise is bounded by $O(d^{-2-\epsilon/8})$.

**Concentration of softmax scores.** Taking $\eta = d^{2+\epsilon/16}$, the updated weights $\mathbf{W}^{(1)} = -\eta \widetilde{\nabla} L(\mathbf{W})$ become

$$w_{j,m}^{(1)} = \frac{2cd^{2+\epsilon/16}}{(m-1)^2} \mathbf{1}_{\{\mathsf{p}[j]=m\}} + O(d^{-\epsilon/16}).$$

In particular, for each $j \neq \mathsf{c}_1[m], \mathsf{c}_2[m]$ the softmax scores satisfy

$$\sigma_j(\boldsymbol{w}_m^{(1)}) = e^{w_{j,m}^{(1)}} / \sum_\alpha e^{w_{\alpha,m}^{(1)}} \leq e^{w_{j,m}^{(1)} - w_{\mathsf{c}_1[m],m}^{(1)}} \leq \exp(-\Omega(d^{\epsilon/16})).$$

As softmax scores must sum to 1, it holds that $\sigma_{\mathsf{c}_1[m]}(\boldsymbol{w}_m^{(1)}) + \sigma_{\mathsf{c}_2[m]}(\boldsymbol{w}_m^{(1)}) \geq 1 - \exp(-\Omega(d^{\epsilon/16}))$ and moreover

$$\frac{\sigma_{\mathsf{c}_1[m]}(\boldsymbol{w}_m^{(1)})}{\sigma_{\mathsf{c}_2[m]}(\boldsymbol{w}_m^{(1)})} = e^{w_{\mathsf{c}_1[m],m}^{(1)} - w_{\mathsf{c}_2[m],m}^{(1)}} \leq \exp(O(d^{-\epsilon/16})) \leq 1 + O(d^{-\epsilon/16})$$

from the inequality $e^t \leq 1 + O(t)$ for small $t > 0$. By symmetry, $\sigma_{c_2[m]}(w_m^{(1)})/\sigma_{c_1[m]}(w_m^{(1)}) \leq 1 + O(d^{-\epsilon/16})$. By simple algebraic manipulation, we can conclude that

$$\frac{1}{2} - O(d^{-\epsilon/16}) \leq \sigma_{c_1[m]}(w_m^{(1)}), \sigma_{c_2[m]}(w_m^{(1)}) \leq \frac{1}{2} + O(d^{-\epsilon/16}).$$

That is, the updated attention layer $\hat{z}_m^{(1)} = \sum_j \sigma_j(w_m^{(1)})x_j$ has learned to take the average of the two child nodes and ignore the remaining input tokens at each step.

**Evaluating the forward pass.** Now to bound the updated prediction loss, we evaluate the error $\|\hat{x}_m^{(1)} - x_m\|_\infty$ of each step of the forward pass for $d + 1 \leq m \leq d + k - 1$. More precisely, define the increasing sequence

$$\epsilon_m = \max_{d < j \leq m} \left\| \hat{x}_j^{(1)} - x_j \right\|_\infty, \quad \epsilon_d = 0.$$

Then

$$\left\| \hat{x}_{c_1[m]}^{(1)} - x_{c_1[m]} \right\|_\infty, \left\| \hat{x}_{c_2[m]}^{(1)} - x_{c_1[m]} \right\|_\infty \leq \epsilon_{c_1[m]}, \epsilon_{c_2[m]} \leq \epsilon_{m-1},$$

and for the intermediate values $\hat{z}_m^{(1)}$ we have

$$\left\| \hat{z}_m^{(1)} - \frac{x_{c_1[m]} + x_{c_2[m]}}{2} \right\|_\infty \leq \left\| \hat{z}_m^{(1)} - \frac{\hat{x}_{c_1[m]}^{(1)} + \hat{x}_{c_2[m]}^{(1)}}{2} \right\|_\infty + \epsilon_{m-1}$$

$$\leq \sum_{p[j] \neq m} \sigma_j(w_m^{(1)}) + \left| \sigma_{c_1[m]}(w_m^{(1)}) - \frac{1}{2} \right| + \left| \sigma_{c_2[m]}(w_m^{(1)}) - \frac{1}{2} \right| + \epsilon_{m-1}$$

$$\leq 2d \exp(-\Omega(d^{\epsilon/16})) + O(d^{-\epsilon/16}) + \epsilon_{m-1}$$

$$\leq C_1 d^{-\epsilon/16} + \epsilon_{m-1},$$

for some constant $C_1 > 0$. Since $\phi$ behaves like a quadratic near $0, \pm 1$, it follows that

$$\epsilon_m = \|\hat{x}_m^{(1)} - x_m\|_\infty = \left\| \phi(\hat{z}_m^{(1)}) - \phi\left( \frac{x_{c_1[m]} + x_{c_2[m]}}{2} \right) \right\|_\infty \leq C_2(C_1 d^{-\epsilon/16} + \epsilon_{m-1})^2$$

for some constant $C_2 > 0$ depending only on $\phi$. Then for sufficiently large $d$, by choosing $C_3$ such that $C_2(C_1 + C_3 d^{-\epsilon/16})^2 \leq C_3$, for $\epsilon_{m-1} \leq C_3 d^{-\epsilon/8}$ it follows that

$$\epsilon_m \leq C_2(C_1 d^{-\epsilon/16} + C_3 d^{-\epsilon/8})^2 \leq C_3 d^{-\epsilon/8},$$

thus $\epsilon_m = O(d^{-\epsilon/8})$ inductively for all $m$. We conclude that $\|\hat{y} - y\|_\infty = \|\hat{x}_{d+k-1}^{(1)} - x_{d+k-1}^{(1)}\|_\infty$ is bounded for all inputs as $O(d^{-\epsilon/8})$. $\qquad\square$

## C    Proof of Theorem 7

**Proof of Lemma 6.** For the iterative generation scheme (3), each $w_{j,m}$ affects $\hat{x}_m$ as well as all nodes $\hat{x}_\alpha$ on higher levels $h[\alpha] > h[m]$ through $\hat{x}_m$. We bound the contribution of each term to the total gradient inductively with respect to the level. Define for each $d < m \leq d + k - 1, 1 \leq j \leq m - 1$ and $0 < \ell \leq v$ the quantity

$$\xi_{j,m,\ell} := \max_{\alpha \leq d_\ell} \left| \frac{\partial \hat{x}_\alpha}{\partial w_{j,m}} \right|.$$

We denote $\kappa := \|\phi'\|_\infty$ for brevity. Clearly $\xi_{j,m,\ell} = 0$ for $\ell < h[m]$ and

$$\xi_{j,m,h[m]} = \left| \frac{\partial \hat{x}_m}{\partial w_{j,m}} \right| \leq \kappa \sigma_j(w_m)|x_j - \hat{z}_m| \leq 2\kappa \sigma_j(w_m).$$

Moreover for any $\alpha$ with $h[\alpha] = \ell > h[m]$, we can bound by the chain rule

$$\left| \frac{\partial \hat{x}_\alpha}{\partial w_{j,m}} \right| \leq |\phi'(\hat{z}_\alpha)| \cdot \left| \sum_{\beta = d_{h[m]}+1}^{d_{\ell-1}} \sigma_\beta(w_\alpha) \frac{\partial \hat{x}_\beta}{\partial w_{j,m}} \right| \leq \kappa \xi_{j,m,\ell-1},$$

yielding the relation $\xi_{j,m,\ell} \leq \kappa\xi_{j,m,\ell-1}$. Iterating, we obtain that $\xi_{j,m,\ell} \leq 2\kappa^{\ell-h[m]+1}\sigma_j(\boldsymbol{w}_m)$. Therefore we can bound the total gradient by the following.

$$
\sum_{\alpha=d+1}^{d+k-1} \|\nabla_{\mathbf{W}}f_\alpha^\times\|^2 = \sum_{\ell=1}^{v}\sum_{\alpha=d_{\ell-1}+1}^{d_\ell} \|\nabla_{\mathbf{W}}f_\alpha^\times\|^2
$$

$$
= \sum_{\ell=1}^{v}\sum_{\alpha=d_{\ell-1}+1}^{d_\ell}\sum_{m=d+1}^{d+k-1}\sum_{j=1}^{m-1}\left|\frac{\partial\hat{x}_\alpha}{\partial w_{j,m}}\right|^2
$$

$$
\leq \sum_{m=d+1}^{d+k-1}\sum_{j=1}^{m-1}\sum_{\ell=1}^{v}(d_\ell - d_{\ell-1})\xi_{j,m,\ell}^2
$$

$$
\leq \sum_{m=d+1}^{d+k-1}\sum_{j=1}^{m-1}\sigma_j(\boldsymbol{w}_m)^2\sum_{\ell=1}^{v}2^{v-\ell}\cdot 4\kappa^{2\ell-2h[m]+2}
$$

$$
\leq 4\sum_{m=d+1}^{d+k-1}2^v\kappa^{-2h[m]+2}\sum_{\ell=1}^{v}\left(\frac{\kappa^2}{2}\right)^\ell \leq \frac{4\kappa^2}{\kappa^2-2}\sum_{m=d+1}^{d+k-1}\kappa^{2v-2h[m]+2}
$$

$$
\leq \frac{4\kappa^2}{\kappa^2-2}\sum_{\ell=1}^{v}(d_\ell - d_{\ell-1})\kappa^{2v-2\ell+2} = \frac{4\kappa^4}{\kappa^2-2}\sum_{\ell=1}^{v}(2\kappa^2)^{v-\ell}
$$

$$
\leq \frac{4\kappa^2}{(\kappa^2-2)(2\kappa^2-1)}(2\kappa^2)^v = O(d^{2\log_2\kappa+1}),
$$

since $2^v = k = O(d)$. $\qquad\square$

We first provide a concentration bound for the augmented data, which we take to hold throughout the proof by conditioning on the high probability event.

**Lemma 10** (concentration of augmented data). *For $n' = \mathrm{poly}(d)$, with probability $1 - e^{-\Omega(\sqrt{d})}$ over random sampling of the augmented data $\boldsymbol{u}_1,\cdots,\boldsymbol{u}_d$, it holds that $\|\boldsymbol{u}_j + \mathbf{1}_{n'}\|_\infty = 2$ for all $1 \leq j \leq d+k-1$ and*

$$
\max_{0\leq\ell\leq v}\left\|\frac{1}{d_\ell}\sum_{j=1}^{d_\ell}\boldsymbol{x}_j^+\right\|_\infty \leq O(d^{-1/4}).
$$

*Proof.* The nodes $x_{d_{\ell-1}+1},\cdots,x_{d_\ell}$ at each level $\ell$ compute independent parities, even though parities at different levels can be correlated. By Hoeffding's inequality and union bounding over coordinates, it follows that

$$
\left\|\sum_{j=d_{\ell-1}+1}^{d_\ell}\boldsymbol{x}_j^+\right\|_\infty \leq \sqrt{2(d_\ell - d_{\ell-1})\log\frac{2n'}{p}} = 2^{\frac{v-\ell+1}{2}}\sqrt{\log\frac{2n'}{p}}
$$

with probability at least $1 - p$. Again union bounding, the above holds for all levels $0 \leq \ell \leq v$ simultaneously with probability at least $1 - vp$, so that

$$
\left\|\frac{1}{d_\ell}\sum_{j=1}^{d_\ell}\boldsymbol{x}_j^+\right\|_\infty \leq \sum_{\ell'=0}^{\ell}\frac{2^{\frac{v-\ell'+1}{2}}}{d}\sqrt{\log\frac{2n'}{p}} \leq 2(\sqrt{2}+1)\sqrt{\frac{1}{d}\log\frac{2n'}{p}} = O(d^{-1/4})
$$

for all $\ell$ if $p = e^{-\sqrt{d}}$. In addition, the probability that $\boldsymbol{u}_j = -\mathbf{1}_{n'}$ for some $j \leq d+k-1$ is bounded by $2d\cdot 2^{-n'} = e^{-\Omega(d)}$; otherwise, at least one entry is equal to 2. $\qquad\square$

Now to prove Theorem 7, we show by induction that with high probability, the weights can be written for constants $C_\ell = \Theta(1)$ as

$$
w_{j,m}^{(t)} = \begin{cases} \mathsf{r}[C_{h[m]}d^{\epsilon/16}] & h[m] \leq t,\ \mathsf{p}[j] = m, \\ -\infty & j > d_{h[m]-1}\text{ or }m \leq d, \\ 0 & \text{otherwise.} \end{cases} \tag{11}
$$

Clearly (11) is satisfied when $t = 0$. Suppose (11) holds at time $t-1$ for $t \geq 1$.

**Evaluating the forward pass.** We first evaluate the forward pass iteration of the transformer up to level $\mathsf{h}[m] \leq t$; fixing $0 < C < \min_{\ell \leq v} C_\ell$, it holds that

$$\sigma_j(\boldsymbol{w}_m^{(t)}) \leq \frac{1}{\exp(w_{\mathsf{c}_1[m],m}^{(t)}) + \exp(w_{\mathsf{c}_2[m],m}^{(t)}) + d_{\mathsf{h}[m]-1} - 2} \leq \exp(-Cd^{\epsilon/16})$$

when $\mathsf{p}[j] \neq m$ and

$$\frac{1 - d\exp(-Cd^{\epsilon/16})}{2} \leq \sigma_{\mathsf{c}_1[m]}(\boldsymbol{w}_m^{(t)}), \sigma_{\mathsf{c}_2[m]}(\boldsymbol{w}_m^{(t)}) \leq \frac{1}{2}.$$

For the augmented tokens, define the increasing per-level error sequence

$$\epsilon_\ell = \max_{d < j \leq d_\ell} \left\| \hat{\boldsymbol{x}}_j^{+(t)} - \boldsymbol{x}_j^+ \right\|_\infty, \quad \epsilon_0 = 0.$$

We recursively bound $\epsilon_\ell$ as before up to $\epsilon_t$; this will simultaneously verify that the filter $\iota$ is not applied for the first $t + 1$ levels since $\|\boldsymbol{u}_j^{(t)} + \mathbf{1}_{n'}\|_\infty \geq 2 - \epsilon_t$ due to Lemma 10.

For each state $\hat{\boldsymbol{z}}_m^{+(t)}$ with $\mathsf{h}[m] = \ell$ we have

$$\left\| \hat{\boldsymbol{z}}_m^{+(t)} - \frac{\boldsymbol{x}_{\mathsf{c}_1[m]}^+ + \boldsymbol{x}_{\mathsf{c}_2[m]}^+}{2} \right\|_\infty$$

$$\leq \sum_{\mathsf{p}[j] \neq m} \sigma_j(\boldsymbol{w}_m^{(t)}) + \left| \sigma_{\mathsf{c}_1[m]}(\boldsymbol{w}_m^{(t)}) - \frac{1}{2} \right| + \left| \sigma_{\mathsf{c}_2[m]}(\boldsymbol{w}_m^{(t)}) - \frac{1}{2} \right| + \epsilon_{\ell-1}$$

$$\leq (2d - 2)\exp(-Cd^{\epsilon/16}) + \epsilon_{\ell-1}.$$

Since $\phi$ behaves like a quadratic near $0, \pm 1$, it follows that

$$\epsilon_\ell \leq C_2((2d-2)\exp(-Cd^{\epsilon/16}) + \epsilon_{\ell-1})^2,$$

and we can inductively verify $\epsilon_\ell \leq \exp(-Cd^{\epsilon/16})$ as well as $\|\hat{\boldsymbol{z}}_m^{+(t)} - \boldsymbol{z}_m^+\|_\infty \leq 2d\exp(-Cd^{\epsilon/16})$ holds for all $\ell \leq t$ for sufficiently large $d$.

On the other hand, for the forward pass for levels $\mathsf{h}[m] > t$ the softmax scores are uniform over $d_{\mathsf{h}[m]-1}$ tokens; moreover, the filter $\iota$ will be applied to all tokens on level $t + 2$ and higher. Indeed, the output of nodes on level $\mathsf{h}[m] = t + 1$ reads

$$\hat{\boldsymbol{x}}_m^{+(t)} = \phi(\hat{\boldsymbol{z}}_m^{+(t)}), \quad \hat{\boldsymbol{z}}_m^{+(t)} = \frac{1}{d_t}(\hat{\boldsymbol{x}}_1^{+(t)} + \cdots + \hat{\boldsymbol{x}}_{d_t}^{+(t)}) = \frac{1}{d_t}(\boldsymbol{x}_1^+ + \cdots + \boldsymbol{x}_{d_t}^+) + O(\epsilon_\ell),$$

so that $\|\hat{\boldsymbol{z}}_m^{+(t)}\|_\infty \leq O(d^{-1/4})$ by Lemma 10. Then

$$\|\hat{\boldsymbol{u}}_m^{(t)} + \mathbf{1}_{n'}\|_\infty \leq C_2\||\hat{\boldsymbol{z}}_m^{+(t)}|^2\| \leq O(d^{-1/2})$$

so that if $O(d^{-1/2}) < \varepsilon_0$, the filter zeroes out the output of each node on level $t + 2$. Then the intermediate states of nodes $x_{m'}$ on level $t + 2$ read

$$\hat{\boldsymbol{z}}_{m'}^{+(t)} = \frac{1}{d_{t+1}}(\hat{\boldsymbol{x}}_1^{+(t)} + \cdots + \hat{\boldsymbol{x}}_{d_t}^{+(t)}) = \frac{d_t}{d_{t+1}}\hat{\boldsymbol{z}}_m^{+(t)},$$

which again activates the filter. Repeating this process for the remaining levels, we conclude that $\|\hat{\boldsymbol{z}}_m^{+(t)}\|_\infty \leq O(d^{-1/4})$ and so $\|\hat{\boldsymbol{x}}_m^{+(t)} + \mathbf{1}_{n+n'}\|_\infty \leq O(d^{-1/2})$ holds simultaneously for all nodes $\mathsf{h}[m] > t$ (and all timesteps $t$ for which (11) is valid).

**Evaluating the updates.** Define $\bar{\boldsymbol{z}}_m^{(t)} = \frac{1}{d_t}\sum_{j=1}^{d_t} \boldsymbol{x}_j$ so that

$$\|\hat{\boldsymbol{z}}_m^{(t)} - \bar{\boldsymbol{z}}_m^{(t)}\|_\infty \leq \frac{1}{d_t}\sum_{j=1}^{d_t}\|\hat{\boldsymbol{x}}_j^{(t)} - \boldsymbol{x}_j\|_\infty \leq \exp(-Cd^{\epsilon/16}).$$

We proceed to evaluate the gradient of $L$ at (11). For the weights $w_{j,m}$ with $\mathsf{h}[m] = t+1$, by isolating the errors from the forward pass we have

$$
\begin{aligned}
\frac{\partial}{\partial w_{j,m}} \left( \frac{1}{2n} \|\hat{\boldsymbol{x}}_m^{(t)} - \boldsymbol{x}_m\|^2 \right) &= \frac{1}{n} \left\langle \phi(\hat{\boldsymbol{z}}_m^{(t)}) - \boldsymbol{x}_m, \frac{\partial \phi(\hat{\boldsymbol{z}}_m^{(t)})}{\partial w_{j,m}} \right\rangle \\
&= \frac{\sigma_j(\boldsymbol{w}_m^{(t)})}{n} \left\langle \phi(\hat{\boldsymbol{z}}_m^{(t)}) - \boldsymbol{x}_m, \phi'(\hat{\boldsymbol{z}}_m^{(t)}), \hat{\boldsymbol{x}}_j^{(t)} - \hat{\boldsymbol{z}}_m^{(t)} \right\rangle \\
&= \frac{1}{n d_t} \left\langle \phi(\bar{\boldsymbol{z}}_m^{(t)}) - \boldsymbol{x}_m, \phi'(\bar{\boldsymbol{z}}_m^{(t)}), \boldsymbol{x}_j - \bar{\boldsymbol{z}}_m^{(t)} \right\rangle \\
&\quad + O\left( \frac{4}{d_t} (1 + \|\phi'\|_\infty + \|\phi''\|_\infty) \|\hat{\boldsymbol{z}}_m^{(t)} - \bar{\boldsymbol{z}}_m^{(t)}\|_\infty \right) \\
&= \frac{1}{n d_t} \left\langle \phi(\bar{\boldsymbol{z}}_m^{(t)}) - \boldsymbol{x}_m, \phi'(\bar{\boldsymbol{z}}_m^{(t)}), \boldsymbol{x}_j - \bar{\boldsymbol{z}}_m^{(t)} \right\rangle + O(\exp(-Cd^{\epsilon/16})).
\end{aligned}
$$

Then the first term is identical to the initial gradient (6) analyzed in the proof of Theorem 5 except for the differences in indices, and from the same computation we obtain the leading term:

$$
\frac{\partial}{\partial w_{j,m}} \left( \frac{1}{2n} \|\hat{\boldsymbol{x}}_m^{(t)} - \boldsymbol{x}_m\|^2 \right) = -\frac{2c}{d_t^2} \mathbf{1}_{\{\mathsf{p}[j]=m\}} + O(d^{-2-\epsilon/8}),
$$

which holds with probability $1 - \exp(-d^{\epsilon/2})$ if $n = \Omega(d^{2+\epsilon})$ under the same setting of Lemma 9.

For all other nodes on level $t+1$ or below, the output does not depend on the weight $w_{j,m}$, so the gradient of the squared error with respect to $w_{j,m}$ is zero. Moreover, all nodes on level $t+2$ or above are zeroed out due to the filter and hence also has zero gradient. Then the oracle error is absorbed into the second term and the update after time $t+1$ with learning rate fixed to $\eta = d^{2+\epsilon/16}\eta_0$ reads

$$
w_{j,m}^{(t)} - \eta \widetilde{\nabla}_{w_{j,m}} L(\mathbf{W}^{(t)}, \mathbf{U}) = 2c\eta_0 \frac{d^{2+\epsilon/16}}{d_{\mathsf{h}[m]-1}^2} \mathbf{1}_{\{\mathsf{p}[j]=m\}} + O(d^{-\epsilon/16}).
$$

By choosing $\eta_0$ such that none of the leading terms lands exactly on a half-integer, we have that for sufficiently large $d$,

$$
w_{\mathsf{c}_1[m],m}^{(t+1)} = w_{\mathsf{c}_2[m],m}^{(t+1)} = \mathsf{r}\left[ 2c\eta_0 \frac{d^{2+\epsilon/16}}{d_{\mathsf{h}[m]-1}^2} \right], \quad w_{j,m}^{(t+1)} = \mathsf{r}[O(d^{-\epsilon/16})] = 0
$$

if $\mathsf{p}[j] \neq m$. We verify that

$$
\frac{c\eta_0}{2} \leq C_\ell = 2c\eta_0 \frac{d^{2+\epsilon/16}}{d_{\ell-1}^2} \leq 2c\eta_0.
$$

Also, the weights $w_{j,m}$ such that $\mathsf{h}[m] \geq t+2$ only affect the nodes that are zeroed out, so that the gradient is also bounded as $O(d^{-\epsilon/16})$ and $w_{j,m}^{(t+1)} = 0$.

It remains to evaluate the gradient signal of weights $w_{j,m}$ with $\mathsf{h}[m] \leq t$, which have already been updated in previous steps. Define the error

$$
\xi_{j,m,\ell} := \max_{1 \leq \alpha \leq d_\ell} \left\| \frac{\partial \hat{\boldsymbol{x}}_\alpha^{(t)}}{\partial w_{j,m}} \right\|_\infty.
$$

This is similar to the error control in the proof of Lemma 6, but we exploit the fact that parities up to level $t$ are solved to obtain a much tighter bound. Let us expand $\phi'(t) = 2c'(1-t) + O((1-t)^2)$ near 1 and $2c'(-1-t) + O((1+t)^2)$ near $-1$ for some positive constant $c'$. Recall

$$
\|\hat{\boldsymbol{z}}_\alpha^{+(t)} - \boldsymbol{z}_\alpha^+\|_\infty \leq 2d\exp(-Cd^{\epsilon/16}), \quad \mathsf{h}[\alpha] \leq t
$$

holds in the forward pass, so each component $\hat{\boldsymbol{z}}_{\alpha,i}^{(t)}$ is $O(\exp(-Cd^{\epsilon/16}))$-close to either of $\pm 1$. It follows that $|\phi'(\hat{\boldsymbol{z}}_{\alpha,i}^{(t)})| = O(\exp(-Cd^{\epsilon/16}))$, so we can bound

$$
\xi_{j,m,\mathsf{h}[m]} = \left\| \frac{\partial \hat{\boldsymbol{x}}_m^{(t)}}{\partial w_{j,m}} \right\|_\infty \leq 2\|\phi'(\hat{\boldsymbol{z}}_m^{(t)})\|_\infty \sigma_j(\boldsymbol{w}_m) \leq O(\exp(-Cd^{\epsilon/16})).
$$

Moreover, for any $\alpha$ on level $\mathsf{h}[\alpha] = \ell$, $\mathsf{h}[m] < \ell \leq t$ the magnitude of the derivative of the output $\hat{\boldsymbol{x}}_\alpha^{(t)}$ can be bounded as

$$\left\| \frac{\partial \hat{\boldsymbol{x}}_\alpha^{(t)}}{\partial w_{j,m}} \right\|_\infty \leq \left\| \phi'(\hat{\boldsymbol{z}}_\alpha^{(t)}) \right\|_\infty \sum_{\beta=1}^{d_{\ell-1}} \sigma_\beta(\boldsymbol{w}_\alpha^{(t)}) \left\| \frac{\partial \hat{\boldsymbol{x}}_\beta^{(t)}}{\partial w_{j,m}} \right\|_\infty \leq O(\exp(-Cd^{\epsilon/16}))\xi_{j,m,\ell-1}.$$

This implies that $\xi_{j,m,t} = \xi_{j,m,t-1} = \cdots = \xi_{j,m,\mathsf{h}[m]} \leq O(\exp(-Cd^{\epsilon/16}))$. Furthermore, for any $\alpha$ on level $\mathsf{h}[\alpha] = t+1$ it holds that

$$\left\| \frac{\partial \hat{\boldsymbol{x}}_\alpha^{(t)}}{\partial w_{j,m}} \right\|_\infty \leq \|\phi'\|_\infty \sum_{\beta=1}^{d_t} \sigma_\beta(\boldsymbol{w}_\alpha^{(t)}) \left\| \frac{\partial \hat{\boldsymbol{x}}_\beta^{(t)}}{\partial w_{j,m}} \right\|_\infty \leq \|\phi'\|_\infty \xi_{j,m,t} \leq O(\exp(-Cd^{\epsilon/16})).$$

Thus we have for all $\alpha$ with $\mathsf{h}[\alpha] \leq t+1$,

$$\frac{\partial}{\partial w_{j,m}} \left( \frac{1}{2n} \|\hat{\boldsymbol{x}}_\alpha^{(t)} - \boldsymbol{x}_\alpha\|^2 \right) = \frac{1}{n} \left\langle \hat{\boldsymbol{x}}_\alpha^{(t)} - \boldsymbol{x}_\alpha, \frac{\partial \hat{\boldsymbol{x}}_\alpha^{(t)}}{\partial w_{j,m}} \right\rangle \leq 2 \left\| \frac{\partial \hat{\boldsymbol{x}}_\alpha^{(t)}}{\partial w_{j,m}} \right\|_\infty = O(\exp(-Cd^{\epsilon/16})),$$

and the nodes on level $t+2$ or above are zeroed out due to the filter. Hence the gradient signal is exponentially small and

$$\widetilde{\nabla}_{w_{j,m}} L(\mathbf{W}^{(t)}, \mathbf{U}) = O(d \exp(-Cd^{\epsilon/16})) + O(d^{-2-\epsilon/8}),$$

so that $w_{j,m}^{(t+1)} = \mathsf{r}[w_{j,m}^{(t)} + O(d^{-\epsilon/16})] = w_{j,m}^{(t)}$. This concludes the proof of (11).

Finally, after time $t = v$ the weights at all levels have been updated, so that repeating the analysis of the forward pass yields that

$$\|\hat{\boldsymbol{y}}_{\text{test}} - \boldsymbol{y}_{\text{test}}\|_\infty = \left\| \hat{\boldsymbol{x}}_{d+k-1}^{(t)} - \boldsymbol{x}_{d+k-1} \right\|_\infty \leq \epsilon_v \leq \exp(-Cd^{\epsilon/16}),$$

as was to be shown. □

## D EXPERIMENTAL DETAILS

For the transformer architecture, the feedforward layer was fixed to the following piecewise quadratic link function:

$$\phi(t) = \begin{cases} -4t^2 - 8t - 3 & t \in [-1, -0.5) \\ 4t^2 - 1 & t \in [-0.5, 0.5) \\ -4t^2 + 8t - 3 & t \in [0.5, 1]. \end{cases}$$

For all CoT models, learning rates were fixed to $\eta = 15, 50, 100$ for $k = 8, 16, 32$. For the direct model, the learning rate was scaled to $0.01\eta$ to ensure stability of training. For the self-consistency model, filtering was done through an equivalent weight-based filter, which checks if any softmax score exceeds a threshold value, here set to $0.4$. Moreover, we found that adding a 10% fraction of the gradient from the prediction loss to that of the CoT loss resulted in more stable training.

Figure 5 shows CoT loss and prediction loss curves for $k = 8, 16, 32$, extending Figure 4. The direct model fails to learn parity in all cases, while CoT with teacher forcing always learns efficiently. For CoT with self-consistency, a similar analysis as in Section 4 can be applied for $k = 8, 16$ with two or three distinct learning stages. We also observe that the basic CoT model manages to fully solve the problem for $k = 8$ (two intermediate levels) but not for $k = 16, 32$ (three and four intermediate levels), indicating that assistance (teacher forcing or self-consistency checking) becomes necessary for more complex tasks.

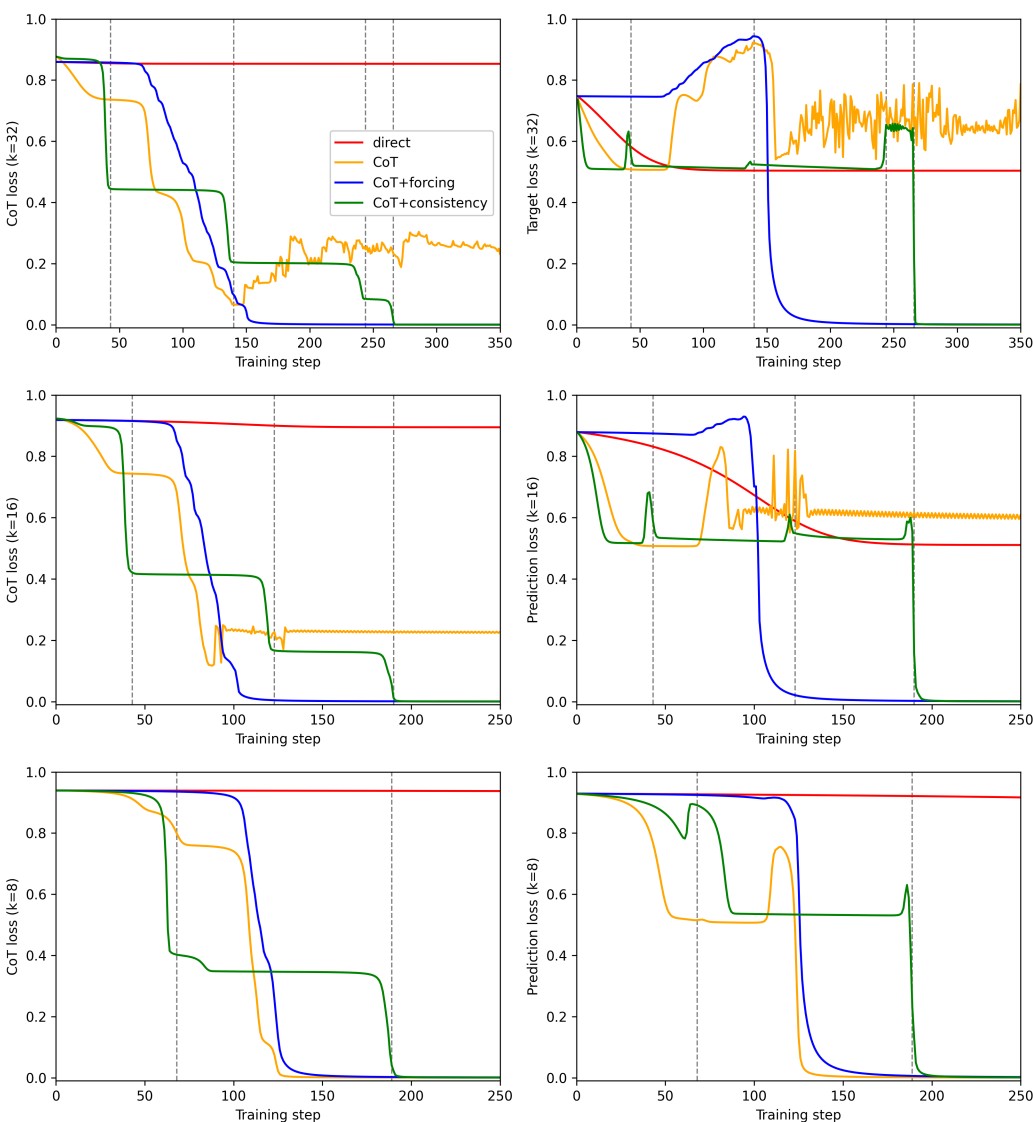

Figure 5: CoT loss (left) and prediction loss (right) curves for the four models when $d = 64$, $k = 32$ (top), $k = 16$ (middle) and $k = 8$ (bottom). For the CoT+consistency model, dashed lines indicate when the filters of each level are deactivated.

