# OpenReview forum: "Transformers Provably Solve Parity Efficiently with Chain of Thought"
_ICLR.cc/2025/Conference — ICLR 2025 Oral_

### Official Review · Reviewer_anZP · 2024-11-01

**Soundness:** 4
**Presentation:** 4
**Contribution:** 3
**Rating:** 8
**Confidence:** 4

**Summary:**

Authors showed that CoT here is beneficial to learn a very hard problem, i.e. bit subset parity. Authors trained a one-layer transformer with a fixed CoT and with a tree based approach to determine the subset and they showed that without CoT it's not possible to learn an algorithm to detect the right bit, _assuming_ data cannot be manipulated freely. They introduced a causal mask to allow less error propagation during CoT.

**Strengths:**

S1. The study is very rigorous, sharing some lights on the importance of CoT on multi hop reasoning.

S2. The introduction of the causal mask is indeed very simple to explain and very intuitive, but also very effective to limit the error compounding. I think this is the main contribution, given teaching forcing is not really useful on this problem.

S3. This work extended previous related work on more realistic setting, offering a clearer picture of why CoT is indeed essential for the given problem.

**Weaknesses:**

W1. Main weakness is a lack of empirical validation. I would love to see just a trained model on various settings (like with and without teacher forcing, parity problem sizes with different $k$ and $d$ values) to show that without teaching forcing the model is indeed able to learn the task.

W2. The claim on the conclusion about finetuning transformer to improve multi step reasoning seems really too strong. In particular, it would be nice to show that it's easy to produce CoT data for all reasoning tasks before claiming this. So I would rephrase that sentence or provide a more detailed discussion on why authors believe their results may extend to other reasoning tasks.

**Questions:**

Q1. Is there any particular reason why one-layer transformer has been chosen? Would the same results (or even stronger) apply to 2 layer transformer?

Q2. Could we apply the same idea to less structured task which lack the hierarchical structure, hence the "easier" teaching forcing?

---

> ### Author Response · Authors · 2024-11-17
>
> Thank you for your detailed review and valuable suggestions, which have helped us greatly in improving the paper! Below are our responses.
>
> **Response to Weaknesses:**
>
> **W1.** We have conducted new experiments supporting our theoretical findings in Section 4 of the updated paper. We set $d=64$, $k=8,16,32$ and implemented four models: direct learning, basic CoT, CoT with teacher forcing, and CoT with self-consistency. These are optimized with a realistic learning schedule (as opposed to the carefully chosen one-step learning rate in our theory), tracking the loss trajctories of both the intermediate states and the final prediction over time. We verified that direct learning and basic CoT fail to solve parity, while CoT with teacher forcing is able to learn efficiently. Moreover, we observed that the masking and filtering procedure for the fourth model helps break the learning into distinct stages, where each level of the parity problem is solved sequentially to 'unlock' the next level. This explicit step-by-step optimization is crucial to arriving at the correct answer. These results confirm that training explicitly for CoT generation can improve performance on multi-step tasks, and that teacher forcing or self-consistency is key to ensuring proper step-by-step learning.
>
> We also varied $k$ in our experiments, and the results have been presented in Appendix D (varying $d$ did not have as much impact). In particular, basic CoT was able to solve parity when $k=8$ (only two intermediate levels) but not when $k=16,32$ (three or four levels), indicating that assistance becomes necessary for more complex tasks. On the other hand, direct optimization was unable to solve parity in all cases. Please see the updated paper for figures and details.
>
> **W2.** Given that we do not study fine-tuning or other practical considerations for LLM optimization, we agree that the conclusion is phrased too strongly and have revised the claim to "Our work takes the first steps towards understanding how CoT can be leveraged to improve multi-step reasoning capability of foundation models." Nevertheless, we hope that our numerical experiments can make the message of our paper more convincing.
>
> **Response to Questions:**
>
> **Q1.** While most existing works on CoT study the expressivity of CoT by proving certain target functions can be expressed by iterating multi-layer transformers, we wanted to shift the focus to optimization and show that CoT-transformers can learn hard tasks such as parity **despite** having a simple one-layer architecture. The solvability of parity with intermediate states by itself is not surprising, for example we can design a gradient-based algorithm which simply checks all ${d\choose 2}$ combinations at each step to solve parity in $O(d^3)$ time, matching Gaussian elimination. Our contribution is to (1) successfully describe the CoT training dynamics of a simple transformer model, and (2) show the emergence of step-by-step reasoning, for which one layer suffices.
>
> Nonetheless, from purely an expressivity point of view, iterating a one-layer transformer $v$ times is a special case of iterating a two-layer transformer $v/2$ times, so we expect more powerful models to exhibit similar/stronger learning capability with shorter CoT lengths. This may result in a tradeoff between less error accumulation due to shorter CoT versus higher difficulty of training due to having more parameters, which seems like an interesting direction for future work. That said, even our one-layer transformer is highly nonlinear and challenging to analyze (softmax attention followed by a nonlinear link function, which is iterated multiple times), so it is likely difficult to obtain a rigorous dynamical analysis for two or more layers.
>
> **Q2.** Our theory is able to handle the tree-like subtask hierarchy of parity (Figure 1) by either linearizing it with teacher forcing (Section 3.2) or optimizing level by level through consistency checks (Section 3.3). We believe this can also give insights to more complex real-world reasoning tasks, which may be expressed as directed graphs connecting certain bits of information to each other via relationships similar to knowledge graphs, e.g. A [CAUSE] B, C [GEQ] D. Through graph traversal methods, a meaningful order could be defined for classes of reasoning tasks for the model to learn step-by-step. Linearization of DAGs can always be performed via Topological Sorting, so teacher forcing should still be possible, but breadth-first traversal may result in shorter CoT which we have seen can be optimized efficiently even without teacher forcing.
>
> We thank you again for your efforts in reviewing our paper and humbly ask to consider raising your score if your concerns have been suitably addressed.

---

> > ### Comment · Reviewer_anZP · 2024-11-17
> >
> > I would like to thank authors for their responses. Before answering to them, do you have an anonymous codebase I can look into?
> >
> > Thanks!

---

> > > ### Author Response · Authors · 2024-11-20
> > >
> > > Since the code is very simple, we have added it as supplementary material. We have also streamlined the code (replacing the augmented data filter by an equivalent weight filter) and added more details to the experimental section in Appendix D. Please inform us if there are further experiments you would like to see!

---

> > > > ### Comment · Reviewer_anZP · 2024-11-21
> > > >
> > > > Thanks for answering W1 so deeply and thanks for changing the wording on W2. I just increased the score to 8, reflecting the high value and high contribution amount the work is providing to the community.

---

### Official Review · Reviewer_5ycE · 2024-11-04

**Soundness:** 3
**Presentation:** 3
**Contribution:** 3
**Rating:** 8
**Confidence:** 3

**Summary:**

This paper provides a theoretical exploration of how transformers develop stepwise reasoning capabilities through recursive task decomposition, using the k-parity problem as a case study. Through formal mathematical proofs and asymptotic complexity analysis, the authors establish three key findings: without intermediate supervision, solving k-parity with finite-precision gradient-based algorithms requires exponentially many steps, extending prior impossibility results to finite-sample settings. In contrast, with teacher forcing—where ground-truth intermediate steps are provided—the model efficiently learns the task in a single gradient update, underscoring the value of explicit supervision for complex reasoning. Additionally, even without teacher forcing, the model achieves accurate reasoning by employing data augmentation to validate intermediate steps, implementing a form of self-consistency check that allows for logarithmic convergence. This research emphasizes theoretical analysis through mathematical proofs and complexity analysis, offering insights into how transformers can perform chain-of-thought reasoning and task decomposition, thereby aligning with recent empirical advances in complex reasoning tasks.

**Strengths:**

1. The paper innovatively uses the k-parity problem to theoretically analyze how transformers develop stepwise reasoning capabilities.
2. This paper introduces a novel hierarchical decomposition of the k-parity problem and designs a corresponding transformer architecture to handle this structure effectively.
3. This paper proposes mechanisms like data augmentation and self-consistency checks, enabling transformers to perform CoT reasoning even in the absence of explicit intermediate supervision.
4. The paper provides clear and logically rigorous analysis for the k-parity problem, offering strong, well-structured insights within this specific context.

**Weaknesses:**

1. The theoretical analysis is heavily focused on the k-parity problem, which, while illustrative, may not extend seamlessly to more complex or varied reasoning tasks. This limits the applicability of the findings to broader transformer applications.
2. The paper lacks empirical experiments to support the theoretical conclusions, which could leave readers questioning the practical effectiveness of the proposed methods in real-world scenarios or on diverse datasets.

**Questions:**

1. For the k-parity problem, each layer’s intermediate state corresponds to a simple “2-parity” check, which is straightforward to define. In more complex tasks, however, defining such intermediate states may not be as intuitive. How could this approach be extended to real-world reasoning tasks where intermediate state decomposition is less clear?
2. The paper uses data augmentation to validate intermediate states in the reasoning process. If applied to practical reasoning tasks, would this data augmentation method require significant adjustments to be effective?
3. Given the proposed architecture, are there potential challenges that might arise in real-world training, particularly concerning stability and computational complexity?

---

> ### Author Response · Authors · 2024-11-17
>
> Thank you for highly evaluating our contribution and sharing insightful suggestions! Below are our responses, including new numerical experiments.
>
> **Response to Weaknesses:**
>
> **W1.** Indeed, our analysis is currently limited to the parity problem. It seems reasonably straightforward to extend the data to more general tokens and 2-parity link function to more general binary operations, to consider the learnability of complex tasks consisting of compounded operations. However, we focused on the parity problem in order to draw a contrast with the negative learning result for parity (Theorem 2). More ambitiously, we could study whether the feedforward layer can learn the binary operation, or consider multiple operations (like addition, multiplication, etc. for arithmetic tasks) and study whether multi-head attention can learn to select the correct one.
>
> **W2.** Taking reviewer feedback into account, we have conducted new experiments supporting our theoretical findings in Section 4 and Appendix D of the updated paper. We set $d=64$, $k=8,16,32$ and implemented four models: direct learning, basic CoT, CoT with teacher forcing, and CoT with self-consistency. These are optimized with a realistic learning schedule (as opposed to the carefully chosen one-step learning rate in our theory), tracking the loss trajctories of both the intermediate states and the final prediction over time. We verified that direct learning and basic CoT fail to solve parity, while CoT with teacher forcing is able to learn efficiently. Moreover, we observed that the masking and filtering procedure for the fourth model helps break the learning into distinct stages, where each level of the parity problem is solved sequentially to 'unlock' the next level. This explicit step-by-step optimization is crucial to arriving at the correct answer. These results confirm that training explicitly for CoT generation can improve performance on multi-step tasks, and that teacher forcing or self-consistency is key to ensuring proper step-by-step learning. Please see the updated paper for figures and details.
>
> **Response to Questions:**
>
> **Q1.** As discussed in W1, our framework could potentially be extended to more general binary operations which combine to give the answer, such as arithmetic-based tasks. Moreover, compared to other CoT works which only consider one-directional task composition, our theory is able to handle the tree-like subtask hierarchy of parity (Figure 1) by either linearizing it with teacher forcing (Section 3.2) or optimizing level by level through consistency checks (Section 3.3). We believe this can also give insights to real-world reasoning tasks, which may be expressed as directed graphs connecting certain bits of information to each other via relationships similar to knowledge graphs, e.g. A [CAUSE] B, C [GEQ] D. Through graph traversal methods, a meaningful order could be defined for classes of reasoning tasks for the model to learn step-by-step. For example, linearization of DAGs can always be performed via Topological Sorting. Alternatively, breadth-first traversal may result in shorter CoT which we have shown can also be optimized efficiently.
>
> **Q2.** In our paper, data augmentation is only used to internally validate each reasoning step; the additional data itself is not necessary. For practical tasks, we believe this can be substituted by any reasonable validation procedure, for example: (1) examining the loss of each specific step and focusing on the first step that is not yet well learned, (2) measuring mutual information between input and output to see if key information is being preserved, (3) using a separate agent classifying each step as valid or invalid. In particular, (1) could also be directly applied to our paper, but we chose the augmentation approach (which does not modify the loss) in order to study how step-by-step naturally arises from a fixed objective.
>
> **Q3.** In our numerical experiments, we found the following considerations to be of practical interest.
>
> (1) learning rate: in Theorems 5,7, we carefully chose a large step size to simplify the analysis. For the experiments, however, a smaller learning rate was used for stable optimization. Nevertheless, the observed training curves match our theoretical results.
>
> (2) stability: CoT assisted with either teacher forcing or the self-consistency procedure resulted in stable training and was able to fully solve parity, aligning with Theorem 5 and 7. On the other hand, training with only end-to-end CoT generation became unstable and failed to make accurate predictions, even after tuning hyperparameters. This supports the use of forcing/validation procedures to control error accumulation.
>
> (3) compute: with 100K 64-bit samples, full-batch gradient descent is unnecessarily memory intensive. Of course, we used full-batch throughout the paper only to contrast with the impossibility result (Theorem 2) and SGD with appropriate batch size should be used in practice.

---

> > ### Comment · Reviewer_5ycE · 2024-11-26
> >
> > Thank you for the response.  I don't have further questions and I am keeping my original score.

---

### Official Review · Reviewer_muTa · 2024-11-04

**Soundness:** 4
**Presentation:** 4
**Contribution:** 4
**Rating:** 10
**Confidence:** 3

**Summary:**

The paper provides a theoretical foundations which explain the benefits of chain-of-though. The authors study a simple setup of k-parity problem with 1-layer Transformer and in this case provide a separation results for transformer trained without intermediate supervision and one trained with teacher forcing. They also provide positive results for the case where the transformer can generate chain-of-thought but is not explicitly supervised with the intermediate results. They show conditions which are required in this case (data augmentation and self-consistency checking).

**Strengths:**

I find the work to be of very high quality and very relevant to the current research in reasoning abilities of language model. The theoretical setup is in my opinion well chosen, very concise and easy to understand.  Even though this is a theoretical paper, the authors are well aware of the current research on the applied side and the questions studied reflect it. I believe it is a strong accept, but I'm giving accept because I'm not sure whether theoretical results of this kind should have spotlight talks at ICLR.

**Weaknesses:**

I was not able to judge how limiting is the setup described in the paper overall, but for example the special masking seems quite artificial but I understand that it is required for the theoretical results. It would be nice if the authors explicitly describe the limitations they are aware of.

**Questions:**

I did not came up with any useful and educated questions. I believe the paper is of high quality and will need to study it in more depth.

---

> ### Author Response · Authors · 2024-11-17
>
> Thank you for your positive assessment of our contribution and valuable comments! Following reviewer suggestions, we have conducted new numerical experiments to complement our theoretical findings and alleviate the main limitations of our paper, which we describe below.
>
> **Response to Weaknesses:**
>
> Our setup is indeed quite specific to the parity problem. While the idea of learning a complex task step-by-step is natural, the studied models (even with one layer) are highly nonlinear and difficult to analyze GD dynamics -- softmax attention followed by a nonlinear link function, which is iterated multiple times. Within this setup, the main limitations are:
>
> (1) A large learning rate, which is carefully calibrated to ensure that every GD update cleanly solves a part of the problem, simplifying the analysis. In practice, a smaller learning rate is needed for stable training.
>
> (2) The masking (and filtering) procedure for the no-forcing setting is quite artificial, as pointed out by the reviewer, and their necessity is not fully justified.
>
> In our numerical experiments detailed in Section 4 and Appendix D of the updated paper, we aimed to address the above two limitations. We set $d=64$, $k=8,16,32$ and implemented four models (direct learning, basic CoT, CoT with teacher forcing, CoT with self-consistency) and optimized them with a relatively small learning rate, tracking the loss trajctories of both the intermediate states and the final prediction over time. We verified that direct learning and basic CoT fail to solve parity, while CoT with teacher forcing is able to learn efficiently. Moreover, we observed that the masking and filtering procedure for the fourth model helps break the learning into distinct stages, where each level of the parity problem is solved sequentially to 'unlock' the next level. This explicit step-by-step optimization is crucial to arriving at the correct answer (in essence, teacher forcing is doing this at all levels simultaneously). Notably, a similar behavior seemed to arise in the basic CoT model as well but failed due to accumulating error, further justifying the use of the filtering mechanism. These results confirm that training explicitly for CoT generation can improve performance on multi-step tasks, and that teacher forcing or self-consistency is key to ensuring proper step-by-step learning. Please see the updated paper for figures and details.
>
> We also mention that ICLR has had oral sessions devoted specifically to theory papers including theoretical and mechanistic analyses of LLMs (2022 Oral 2, 2023 Oral 6 Track 1, 2024 Oral 2D), so if the reviewer feels like recommending our paper for a talk, please go ahead!

---

> > ### Comment · Reviewer_muTa · 2024-11-26
> >
> > Thank you for the detailed response. You have convinced me to increase the score and recommend the paper for a talk.

---

### Meta-Review · Area_Chair_c4ZC · 2024-12-20

**Metareview:**

The submission studies a simple setup of k-parity problem with 1-layer Transformer and provides a separation results for transformer trained without intermediate supervision and one trained with teacher forcing, thereby showing the importance of chain-of-thought.

+ The paper is clearly written.
+ The analysis is thorough.

- Initially, there were some concerns about the lack of empirical evaluation.

**Additional Comments On Reviewer Discussion:**

The responses provided by the authors addressed the issues that were raised in the initial reviews. All the reviewers strongly recommend acceptance.

---

### Decision · Program_Chairs · 2025-01-22

Accept (Oral)